# Transforming Growth Factor-β1 Selectively Recruits microRNAs to the RNA-Induced Silencing Complex and Degrades CFTR mRNA under Permissive Conditions in Human Bronchial Epithelial Cells

**DOI:** 10.3390/ijms20194933

**Published:** 2019-10-05

**Authors:** Nilay Mitash, Fangping Mu, Joshua E. Donovan, Michael M. Myerburg, Sarangarajan Ranganathan, Catherine M. Greene, Agnieszka Swiatecka-Urban

**Affiliations:** 1Department of Pediatrics, UPMC Children’s Hospital of Pittsburgh, University of Pittsburgh School of Medicine, Pittsburgh, PA 15224 USA; jed123@pitt.edu; 2Center for Research Computing, University of Pittsburgh, Pittsburgh, PA 15260, USA; fangping@pitt.edu; 3Division of Pulmonary, Allergy and Critical Care Medicine, University of Pittsburgh School of Medicine, Pittsburgh, PA 15122, USA; myerburgm@upmc.edu; 4Department of Pathology, UPMC Children’s Hospital of Pittsburgh, University of Pittsburgh School of Medicine, Pittsburgh, PA 15224, USA; Sarangarajan.Ranganathan@chp.edu; 5Department of Clinical Microbiology, Royal College of Surgeons in Ireland, Dublin 9, Ireland; CMGreene@rcsi.ie

**Keywords:** cystic fibrosis, CFTR, TGF-β1, microRNA, primary human bronchial epithelial cells

## Abstract

Mutations in the Cystic Fibrosis Transmembrane Conductance Regulator (*CFTR*) gene lead to cystic fibrosis (CF). The most common mutation F508del inhibits folding and processing of CFTR protein. FDA-approved correctors rescue the biosynthetic processing of F508del-CFTR protein, while potentiators improve the rescued CFTR channel function. Transforming growth factor (TGF-β1), overexpressed in many CF patients, blocks corrector/potentiator rescue by inhibiting CFTR mRNA in vitro. Increased TGF-β1 signaling and acquired CFTR dysfunction are present in other lung diseases. To study the mechanism of TGF-β1 repression of CFTR, we used molecular, biochemical, and functional approaches in primary human bronchial epithelial cells from over 50 donors. TGF-β1 destabilized CFTR mRNA in cells from lungs with chronic disease, including CF, and impaired F508del-CFTR rescue by new-generation correctors. TGF-β1 increased the active pool of selected micro(mi)RNAs validated as CFTR inhibitors, recruiting them to the RNA-induced silencing complex (RISC). Expression of F508del-CFTR globally modulated TGF-β1-induced changes in the miRNA landscape, creating a permissive environment required for degradation of F508del-CFTR mRNA. In conclusion, TGF-β1 may impede the full benefit of corrector/potentiator therapy in CF patients. Studying miRNA recruitment to RISC under disease-specific conditions may help to better characterize the miRNAs utilized by TGF-β1 to destabilize CFTR mRNA.

## 1. Introduction

Cystic Fibrosis Transmembrane Conductance Regulator (CFTR) is a cAMP-activated anion channel essential to airway surface liquid (ASL) homeostasis, mucus hydration, and mucociliary clearance [1]. Mutations in the *CFTR* gene lead to cystic fibrosis (CF). The classic severe CF phenotype with recurrent pulmonary exacerbations and pancreatic insufficiency develops when CFTR channel function is less than 1% of normal, progressive pulmonary disease develops when function is less than 4.5%, while at least 10% function is needed to alleviate the most severe symptoms [2]. Deletion of the codon for phenylalanine at position 508 (F508del) is the most common CF-associated mutation, present in almost 90% of CF patients. It causes defects at the level of CFTR mRNA, protein, and anion channel function. First, the deletion alters the F508del-CFTR mRNA structure and reduces translation [3]. Second, it arrests folding and processing of F508del-CFTR, leading to subsequent degradation of most of the protein [4]. F508del-CFTR is retained in the endoplasmic reticulum (ER) in an immature, partially glycosylated form. Partial rescue of the processing defect in vitro allows exit of some of F508del-CFTR from the endoplasmic reticulum (ER), maturation while passing through the Golgi complex, and trafficking to the cell membrane. Third, rescued F508del-CFTR has severely impaired channel function and reduced membrane residence [5,6]. Kalydeco (Ivacaftor; VX-770) is a potentiator that increases the open probability of membrane-resident CFTR channels and is approved by the U.S. Food and Drug Administration (FDA) for individuals with responsive gating mutations (~15% of CF patients) [7,8,9]. Improvement of lung function in these patients was associated with rescue of CFTR activity to 35%–40% of normal, corresponding with the mean absolute improvement in the percentage of the predicted forced expiratory volume in one second (FEV1) of 10%. Although VX-770 had no effect for F508del patients, its development was a major breakthrough, since it was the proof-of-concept that small-molecule therapy may improve CFTR function [10]. 

Lumacaftor (VX-809) and tezacaftor (VX-661) are FDA-approved CFTR correctors that, when combined with VX-770 (dual therapy), modestly reduced exacerbation rates and respiratory symptoms [11,12,13]. The newest correctors, VX-659 and VX-445, have recently demonstrated profound clinical promise because of an additive benefit when combined with the dual therapy with VX-661/770. In the first phase 2 trial, the VX-659/661/770 triple-therapy improved lung function and significantly increased the primary end-point of percent predicted of FEV1 in F508del homozygous patients by an average of 9.7% [14]. Similar results were reported in the second phase 2 trial, examining triple therapy with VX-445/661/770 [15]. Both new-generation therapies improved sweat Cl^−^ concentrations and patient-reported outcomes. Whether these effects would be sustained, decrease exacerbations, and lead to other meaningful outcomes will be answered by on-going phase 3 clinical trials. 

Predicting the future of CF lung disease in the era of new-generation modulators is difficult, since many internal and external factors influence disease severity [16]. For example, non-CFTR modifier genes, including *TGF-β1*, affect the severity of the CF lung disease in F508del homozygous individuals [17,18,19,20]. Data published by several research groups, including ours, suggest that inhibition of F508del-CFTR transcript by TGF-β1 may prevent the full beneficial effects of corrector VX-809 [21,22,23]. However, the effects of TGF-β1 on F508del-CFTR corrected by the new-generation therapies are unknown. 

Little is known as to whether TGF-β1 inhibits F508del-CFTR mRNA by transcriptional or post-transcriptional mechanisms. Post-transcriptional regulation by miRNAs, as effectors of TGF-β1, has been suggested because specific oligonucleotide against an experimentally validated CFTR inhibitor miRNA-145-5p (referred to as miR-145 in the manuscript) partially prevented TGF-β1 inhibition of the corrector-rescued F508del-CFTR in primary differentiated human bronchial epithelial (HBE) cells [24]. Initial work suggested that miRNAs primarily inhibit protein translation, but the current model indicates that miRNAs lead to degradation of the target mRNA [25]. If miRNAs are the mediators, it is expected that treatment with TGF-β1 should decrease the CFTR mRNA level by accelerating its degradation. Yet, the degradation of F508del-CFTR mRNA after TGF-β1 treatment has not been demonstrated. 

The miRNA-mediated inhibitory gene regulation involves extensive processing, during which a miRNA is recruited to RISC and bound to the target mRNA recognition sequence(s), usually in the mRNA 3’-untranslated region (UTR) [25]. Assembly of the miRNA-RISC-mRNA functional complex leads to degradation of the target mRNA and thus, the RISC-associated pool of miRNA is considered active and relevant to the inhibitory gene regulation [26]. We have previously shown increased miR-145 level in bronchial brushings from F508del CF patients, compared to non-CF controls [27]. An increased miR-145 level was also reported in primary nasal epithelial cells from CF patients with different mutations, including F508del [28]. Yet, it is unknown whether RISC recruitment of miR-145 or other miRNAs is increased in the F508del environment. 

Cross-talk between TGF-β1 signaling and the miRNA machinery is necessary for homeostasis, while its dysregulation leads to disease [29,30,31]. Transcriptional regulation of miR-145 expression by canonical TGF-β1 signaling was previously shown in mesenchymal stem cells, vascular smooth muscle cells (SMCs), and prostate cancer cells [30,32,33,34]. In chronic obstructive pulmonary disease (COPD), miR-145 mediates the release of pro-inflammatory cytokines from SMCs under TGF-β1 control [35,36]. Dysregulation of homeostasis in the lung microenvironment by aberrant TGF-β1-miRNA cross-talk also contributes to the pathogenesis of idiopathic pulmonary fibrosis (IPF) [37,38]. However, little is known about dysregulation of the TGF-β1-miRNA crosstalk in CF. 

Besides miR-145, several other miRNAs have been experimentally validated as CFTR inhibitors [27,34,39,40,41]. While there is no direct evidence that TGF-β1 affects the miRNA-RISC-mRNA complex, recent data demonstrate that an inhibitor of protein–nucleic acid interaction partially rescued TGF-β1 repression of CFTR mRNA in non-CF HBE cells [42]. 

To answer these questions and validate the predictions, studies were conducted in native human bronchial tissue, as well as primary and immortalized human bronchial epithelial cell models. Here, we show that chronic lung disease, including CF, creates a permissive environment for TGF-β1-miRNA cross-talk, leading to selective recruitment of validated CFTR inhibitors to RISC and decay of CFTR mRNA. As a result, TGF-β1 blocks the F508del-CFTR rescue by the new-generation correctors. Our data increase understanding of the role of TGF-β1 in CF and other forms of chronic lung disease and may help to develop future strategies to achieve the full benefit of the transformative CFTR therapies in CF patients homozygous for F508del.

## 2. Results

### 2.1. TGF-β1 Inhibits Rescue of F508del-CFTR and ASL Volume by the New-Generation Correctors

Since the discovery of VX-809, newer, more efficient correctors have emerged, but it is unknown whether TGF-β1 interferes with their rescuing ability. For example, the combined use of corrector C18 and CFFT-002 (C002) had a superior effect on F508del-CFTR rescue in vitro, compared to a single administration of either corrector or other small molecules [43]. Thus, we examined the effect of TGF-β1 in the presence of the correctors. HBE cells—homozygous for the F508del mutation—were cultured in air–liquid interface (ALI) for 6 weeks to full differentiation and correctors C18 and C002 (5 μM each) or vehicle control were added to the basolateral medium for 48 h, and TGF-β1 (15 ng/mL) or vehicle control for 24 h before cell lysis. CFTR protein abundance was examined by Western blotting (WB) and quantified by densitometry. HBE cells from COPD lungs were used as an additional control for CFTR protein abundance. The combined use of C18 and C002 increased the abundance of the partially glycosylated CFTR band B and fully glycosylated band C in F508del HBE cells and TGF-β1 reduced the abundance of corrector-rescued CFTR band B and C (Figure 1A,B).

Next, we examined TGF-β1 effects on the corrector C18/C002 rescue of the CFTR-mediated short circuit current (*I*sc). F508del HBE cells were treated with correctors C18/C002 and TGF-β1 or vehicle. In the Ussing chamber, amiloride was used first to inhibit epithelial sodium channel (ENaC). Forskolin, a protein kinase A (PKA) activator, and isobutylmethylxanthine (IBMX), a non-specific phosphodiesterase inhibitor, were used to increase and maintain cAMP levels, respectively, and potentiator VX-770 was used to potentiate the open probability of the corrector-rescued F508del-CFTR anion channel, while CFTR_inh_-172 was used to inhibit it [44]. Correctors C18/C002 rescued CFTR-mediated Isc, while TGF-β1 blocked the functional rescue of CFTR and inhibited the ENaC-mediated Isc (Figure 1C–E). A 48 h treatment with the FDA-approved corrector VX-661 (5 μM) increased CFTR-mediated Isc in F508del HBE cells, although the effect was inferior to the combined use of C18 and C002, and TGF-β1 inhibited CFTR rescue by VX-661 (Figure 1F,G versus Figure 1C,D). Because of the superior effect of the combined use of corrector C18 and C002, the pair was used in subsequent experiments with F508del HBE cells. 

ENaC reabsorbs Na^+^ and water, and together with CFTR, creates a balance to maintain a sufficient volume of the ASL necessary for mucociliary clearance [45]. ENaC is negatively regulated by CFTR [46,47]. In the absence of CFTR function and reduced Cl^−^ secretion in CF, unopposed ENaC activity results in Na^+^ hyperabsorption that impairs ASL volume regulation. Although it has been shown that TGF-β1 inhibits ENaC-mediated Isc, as summarized in Figure 1E, the effect of inhibiting both channels on ASL volume regulation is not well understood [48,49]. Thus, we examined how TGF-β1 affects the corrector-rescue of ASL in F508del HBE cells by meniscus scanning [50]. ASL volume in ALI cultures was measured by scanning the apical meniscus before treatment. Cells were treated with correctors C18/C002 and scanned 24 h later. Subsequently, cells were treated with TGF-β1 or vehicle in the presence of correctors C18/C002 and scanned again 24 h later. In the presence of vehicle control, correctors increased the ASL volume, while TGF-β1 completely blocked it (Figure 1H). These data show the net inhibitory effect of TGF-β1 on the corrector-rescued ASL volume. Together, these data demonstrate that TGF-β1 acts upstream of different small molecules, limiting the full benefit of the corrector/potentiator rescue of F508del-CFTR and ASL volume.

### 2.2. TGF-β1 Facilitates Degradation of CFTR mRNA 

We examined whether TGF-β1 represses F508del-CFTR mRNA by transcriptional and/or post-transcriptional mechanisms. First, we studied TGF-β1 effects on CFTR promoter activity using the firefly luciferase construct pCFTR-pLuc [51,52]. Luciferase expression is driven by a 1 kb segment of the human CFTR promoter containing the start codon, as well as transcription factor binding sites, including cAMP-responsive element (CRE) and variant CRE activated by PKA. Human embryonic kidney (HEK) cells, known for the absence of endogenous CFTR expression, were transfected with pCFTR-pLuc construct, or vector control and 24 h later, cells were treated with forskolin, TGF-β1 or vehicle control. Luciferase activity, measured by chemiluminescence, was increased by pCFTR-pLuc expression, compared to vector control (Figure 2A). In the pCFTR-pLuc expressing cells, forskolin increased the luciferase activity and TGF-β1 had no effect. These data demonstrate that CFTR promoter activity, regulated by CRE in a PKA-dependent manner, is insensitive to TGF-β1. Next, we studied TGF-β1 effects on the stability of F508del-CFTR mRNA. F508del HBE cells were treated with actinomycin D (ActD) to inhibit new mRNA transcription, and TGF-β1 or vehicle control was added for 4, 8, or 24 h. The calculated mRNA half-life was 21.10 h and 13.66 h for te vehicle and TGF-β1-treated cells, respectively (Figure 2B). These data provide the first direct evidence that TGF-β1 inhibits CFTR mRNA post-transcriptionally, by increasing its decay. 

### 2.3. Native Bronchial Epithelia from Lungs WITH Chronic Disease Express Higher mir-145 Levels 

Increased decay of CFTR mRNA focused our attention on miRNAs as TGF-β1 mediators. miR-145 has been experimentally validated in vitro as a CFTR inhibitor and it recently emerged as a possible mediator of TGF-β1 repression of CFTR [24,27,39]. Increased miR-145 levels have been observed in bronchial brushings from F508del homozygous patients, compared to controls [27]. Thus, we first characterized the endogenous expression of miR-145 in human bronchial tissue. miR-145 is highly expressed in SMCs and has a well-documented role in airway pathophysiology, including the release of pro-inflammatory cytokines from SMCs in COPD patients, where its expression is controlled by TGF-β1 [35,36]. Thus, SMCs and COPD bronchial epithelia served as positive controls. Evaluation by in situ hybridization (ISH) demonstrated high miR-145 expression in the COPD bronchial epithelia and undetectable expression in epithelia without chronic lung disease (control; Figure 3A and Table 1). F508del homozygous bronchial epithelia expressed elevated levels of miR-145, compared to controls. Examination of epithelia from an IPF lung showed miR-145 expression similar to COPD (Table 1). These data suggest that miR-145 expression is elevated in different forms of chronic lung disease. The intensity of the TGF-β1 pathway activation can be controlled by expression of TGF-β receptor (TβR)-I and TβR-II. Examination of the above-mentioned bronchial tissues by immunohistochemistry (IHC) showed similar levels of TβR-I and TβR-II in all tissues, suggesting that miR-145 levels are not controlled by modulating expression of TβR-I or TβR-II (Figure 3B and Table 1).

### 2.4. Upregulation of the Total Cellular miR-145 Levels by TGF-β1 Does Not Correlate with Repression of CFTR mRNA 

Next, we examined how TGF-β1 affects the expression of miR-145 and CFTR mRNA in HBE cells from lungs homozygous for F508del, COPD or IPF (other disease), or control group from lungs without known chronic lung disease. Cells were treated with TGF-β1 or vehicle for 24 h and the levels of miR-145 and CFTR mRNA were examined by qRT-PCR. In the presence of vehicle, miR-145 expression was higher in the other disease group, compared to the F508del or control group (Figure 4A). These data are consistent with the increased baseline of miR-145 expression in native bronchial epithelial samples from COPD and IPF patients (Figure 3A and Table 1). These data show that HBE cells retain chronic lung disease characteristics of the donor lung with respect to miRNA expression and hence validate the cells as an adequate model for our studies. TGF-β1 increased miR-145 expression in HBE cells in all groups (Figure 4A). As shown in Figure 4B, the expression was increased by > 1 Log2 fold change (FC) in all cells lines in the control group (*n* = 16/16) and in the majority of cell lines in the F508del and other disease group (*n* = 13/14 and 8/11, respectively). 

The baseline CFTR mRNA levels were similar across all HBE cell lines in the presence of vehicle, while TGF-β1 inhibited its levels in the F508del and other disease cells but not in the control cells (Figure 4C). TGF-β1 decreased CFTR mRNA level in a significantly higher number of cell lines in the F508del and other disease group, compared to the control group (Figure 4D). Paired analysis showed no correlation between miR-145 and CFTR mRNA levels after TGF-β1 treatment (Figure 4E). These results show that the TGF-β1-induced upregulation of miR-145 expression cannot explain the inhibition of CFTR mRNA in HBE cells. 

To examine more thoroughly the relationship between the expression of miR-145 and CFTR mRNA, we increased miR-145 expression using a synthetic analog of miR-145. Control HBE cells were treated with 5 nM miRCURY LNA™ Power Mimic hsa-miR-145-5p (miR-145 analog), or miRCURY LNA™ scrambled control (control analog) for 48–72 h. Despite increasing total cellular miR-145 levels, CFTR mRNA expression and CFTR-mediated Isc remained unchanged (Figure 5A–D). Parallel experiments performed in F508del HBE cells demonstrated that the miR-145 analog did not inhibit the corrector-mediated rescue of F508del-CFTR Isc (Figure 5E–G). Thus, upregulating miR-145 levels by a synthetic analog did not decrease CFTR mRNA level or its function in either control or F508del HBE cells.

### 2.5. Decreasing miR-145 Expression Does Not Increase CFTR mRNA Level

Next, we examined the effect of decreasing expression of endogenous miR-145 on CFTR mRNA levels. Expression of miR-145 is controlled by the canonical TGF-β1 signaling pathway that commences with activation of receptor regulated (R)-Smads, Smad2 and Smad3 by TβR-I, followed by their nuclear translocation [30]. F508del HBE cells were transduced with shRNA directed against Smad2 (shSmad2), Smad3 (shSmad3), or scrambled control (shControl) and cells were cultured in ALI for 6 weeks. shSmad2 decreased Smad2 protein abundance by more than 80% and had no significant effect on Smad3, while shSmad3 decreased Smad3 abundance by ~45% without significantly affecting Smad2 (Figure 6A,B). Knockdown (KD) of Smad2 or Smad3 significantly decreased miR-145 expression, but had no effect on CFTR mRNA level (Figure 6C,D). These results confirm that canonical TGF-β1 signaling controls miR-145 expression in HBE cells, but lowering the endogenous expression of miR-145 does not increase CFTR mRNA level in F508del HBE cells.

### 2.6. Anti-miR-145 Oligonucleotide Completely Blocks TGF-β1 Upregulation of miR-145 Expression but Only Partially Reverses TGF-β1 Inhibitory Effects on F508del-CFTR in Select HBE Cell Lines

Next, we examined the effects of preventing TGF-β1 upregulation of miR-145 on CFTR mRNA in F508del HBE cells. We used specific antisense oligonucleotides to sequester miR-145 and prevent downstream effects [53]. ALI cultures of F508del HBE cells were pretreated with 50 nM LNA-antisense probe specific for miR-145-5p miRCURY LNA™ Power Inhibitor hsa-miR-145-5p (Inh-145), or scrambled miRCURY LNA™ control (Inh-Control) for 8 days. Correctors C18/C002 were added 48 h before experiments and TGF-β1 or vehicle was added 24 h before the experiment. TGF-β1 increased miR-145 levels in all cells treated with Inh-Control, while its effect was completely blocked by Inh-145 (Figure 7A). In the presence of Inh-Control, CFTR mRNA expression was reduced by TGF-β1 in all examined F508del HBE cell lines, while Inh-145 had no significant effect, partially preventing the repression in four out of seven examined cell lines (Figure 7B). 

Next, we examined the functional effect of miR-145 on corrector-mediated rescue of F508del-CFTR Isc. To study the effect, we chose two F508del HBE cell lines in which miR-145 partially reversed the TGF-β1 repression of CFTR mRNA (Figure 7B). Cells were treated with correctors C18/C002, Inh-Control, or Inh-145 and TGF-β1 or vehicle. TGF-β1 blocked the corrector-rescue of CFTR Isc in the presence of Inh-Control, while Inh-145 partially reversed the effect (Figure 7C,D). The effect of Inh-145 was specific for F508del-CFTR because it did not reverse the TGF-β1 inhibition of the ENaC-mediated Isc (Figure 7C–E). Taken together, a highly efficient anti-miR-145 oligonucleotide was insufficient to reverse TGF-β1 repression of CFTR mRNA and function despite completely blocking TGF-β1 upregulation of miR-145. These data strongly suggest that the validated CFTR inhibitor miR-145 is not the only TGF-β1 mediator and/or the inhibitory effect of miR-145 does not depend on the total cellular miR-145 levels.

### 2.7. F508del-CFTR Modulates TGF-β1-Induced Changes in miRNA Landscape 

We used small RNA-seq to examine TGF-β1 effects on the entire miRNA landscape. Experiments were performed in polarized human bronchial epithelial cell models CFBE41o- expressing WT- or F508del-CFTR (WT-CFBE and F508del-CFBE cells, respectively). Cells were treated with TGF-β1 or vehicle for 24 h, total RNA was isolated and used with QIAseq^TM^ miRNA Library Kit (Qiagen), and analyzed on the Illumina platform. The miRNA sequencing data were analyzed using the GeneGlobe Data Analysis bioinformatics pipeline from Qiagen. The F508del variant significantly affected the TGF-β1-induced changes in the miRNA landscape (Table 2 and Figure 8A,B). 

As shown in Figure 8A,B, TGF-β1 changed the expression of miRNAs with binding sites in the human CFTR 3’UTR predicted by TargetScan Release 7.2, Diana-TarBase v.8, miRBase, and/or miRmap bioinformatics tools [54,55,56,57]. In WT-CFBE cells, TGF-β1 decreased expression of 16 and increased 28 miRNAs with binding sites in CFTR 3’UTR (Figure 8A,C and Appendix A). In F508del-CFBE cells, TGF-β1 decreased expression of 27 and increased 23 miRNAs predicted to target CFTR 3’UTR (Figure 8B,C and Appendix A). 

### 2.8. TGF-β1 Selectively Recruits Validated CFTR Inhibitors to RISC 

Several lines of evidence, summarized in Figure 4, Figure 5, Figure 6, Figure 7 and Figure 8, showed that total cellular miR-145 expression does not correlate with its inhibitory potential for CFTR mRNA. The inhibitory gene regulation by miRNA requires its recruitment to RISC [26]. Thus, in the next set of experiments, we examined whether TGF-β1 affected recruitment of miR-145 and other CFTR inhibitors to RISC using the RISC immunoprecipitation (IP) assay (RIP). ALI cultures of WT- or F508del-CFBE cells were treated with TGF-β1 or vehicle for 24 h. Cells were lysed, Argonaute (Ago)2 protein—a major component of RISC—was immunoprecipitated, detected by WB, and analyzed by densitometry. Subsequently, the co-IP of miR-145 with Ago2 was measured by q-RT-PCR. The Ago2 protein abundance and the Ago IP efficiency were similar in the presence of vehicle or TGF-β1 in both cell lines (Figure 9A–C). miR-145 was present in the IP complex with Ago2 in the WT- and F508del-CFBE cells (Figure 9D). TGF-β1 increased miR-145 co-IP with Ago2 in both cell lines despite having no significant effect on the total cellular miR-145 level (Figure 9D versus Figure 8C and Appendix A). Next, we examined whether TGF-β1 recruits additional CFTR inhibitors to RISC. miR-143-5p (miR-143) is transcribed together with miR-145 from chromosome 6 and we recently experimentally validated it as a CFTR inhibitor [41]. Although upregulation of miR-143 by TGF-β1 did not reach statistical significance in either WT- or F508del-CFBE cells in the small RNA-seq experiments (Figure 8), miR-143 was present in the Ago2 IP complex in the WT- and F508del-CFBE cells and TGF-β1 increased its co-IP with Ago2 in both cell lines (Figure 9E). These data indicate that, similar to miR-145, miR-143 may be recruited to RISC by TGF-β1 to co-mediate CFTR mRNA repression. 

Since the TGF-β1 recruitment of miR-143 and miR-145 to RISC did not require increasing their total cellular levels, we examined the RISC recruitment of a miRNA whose total cellular level was increased by TGF-β1. Among the predicted and uniquely upregulated miRNAs in F508del-CFBE cells, miR-154-5p (miR-154) was the only top candidate previously associated with lung disease (Figure 8C and Appendix A) [55,57,58,59]. Although TGF-β1 specifically increased miR-154 expression in F508del-CFBE cells, it did not increase its association with Ago2 (Figure 8C versus Figure 9F). Based on these results, we conclude that despite being upregulated by TGF-β1 in F508del-CFBE cells, the predicted CFTR inhibitor miR-154 does not mediate TGF-β1 repression of CFTR mRNA. Taken together, our data add another line of evidence that TGF-β1 recruitment of a miRNA to RISC does not correlate with the TGF-β1 effect on the total cellular miRNA level. The search for additional TGF-β1 mediator(s) of CFTR repression should focus on the active, RISC-associated miRNA fraction rather than total miRNA expression. Using this approach, we show that TGF-β1 may selectively recruit miR-145 and miR-143 to co-mediate repression of CFTR mRNA, although future studies may identify additional mediators.

## 3. Discussion

This study shows that TGF-β1 inhibits CFTR mRNA in human bronchial epithelial cells from lungs with chronic disease, including CF, COPD, and IPF. TGF-β1 increases the active pool of validated CFTR inhibitors, miR-143 and miR-145 to mediate degradation of CFTR mRNA. In turn, reduced CFTR mRNA level compromises the corrector/potentiator rescue of F508del-CFTR mRNA and channel function. Our data suggest that TGF-β1 may impede the full benefit of the novel transformative CFTR therapies in CF patients with the F508del mutation. Moreover, TGF-β1 may contribute to the acquired CFTR dysfunction in COPD and IPF through miRNA-mediated mechanisms. The conclusions, summarized in Figure 10, are supported by several lines of evidence acquired by complementary approaches in relevant cell models.

First, we show that TGF-β1 inhibits rescue of F508del-CFTR at the level of protein and channel function and prevents regulation of ASL volume by the newer generation correctors (Figure 1). These data expand previous studies showing that TGF-β1 blocks the corrector VX-809-mediated functional rescue of F508del-CFTR in HBE cells [22,23]. The contribution of ENaC to ASL dehydration in CF is still debated. We show that TGF-β1 has a net inhibitory effect on the corrector-rescued ASL volume in F508del HBE cells by repressing CFTR and ENaC, two channels with opposite effect on ASL hydration (Figure 1). 

Second, we provide direct evidence that TGF-β1 does not inhibit activity of the minimal CFTR promoter, but facilitates degradation of CFTR mRNA (Figure 2). These data extend previous work showing that TGF-β1 does not affect transcription initiation from the CFTR promoter [42]. Data showing that TGF-β1 targets CFTR post-transcriptionally, while CFTR correctors act post-translationally, explain why TGF-β1 inhibits rescue of F508del-CFTR by different correctors, and may impede the rescuing action of future therapies targeting CFTR downstream of TGF-β1.

Third, we provide novel information that CF, COPD, and IPF create an environment needed for TGF-β1 to degrade CFTR mRNA (Figure 4C). The TGF-β1 signaling pathway plays a critical role in lung organogenesis and maintenance of homeostasis, while it is dysregulated during chronic lung disease [60]. In CF patients, the TGF-β1 ligand is upregulated during Pseudomonas aeruginosa infection or colonization, poor nutritional status, and tobacco smoke exposure [61,62,63]. The importance of two allelic variants of the TGF-β1 gene, present in 40% of F508del homozygous patients, is emphasized by their association with more severe lung disease and exacerbation of the pernicious effect of second-hand smoking in CF [18,20,64]. The TGF-β1 pathway has been also implicated in acquired CFTR dysfunction by cigarette smoke exposure in COPD and acute lung injury [21,65,66,67,68,69]. Several studies reported inhibition of CFTR transcript or channel function by TGF-β1 in different airway epithelial cell models [21,22,23,24,70,71]. We have previously reported an inhibitory effect of TGF-β1 on CFTR in F508del HBE cells and the majority of the non-CF controls that included lungs without chronic disease, as well as COPD and IPF [22]. In the present study, examination of more than 50 HBE cell lines, grouped by disease background, allowed to extend our previous findings and demonstrate that TGF-β1 inhibits CFTR in HBE cells from CF, COPD, and IPF lungs, compared to controls without chronic disease (Figure 4C). At the present time, we do not know how the three conditions enable TGF-β1 to repress CFTR. 

Fourth, we show that total cellular levels of miR-145, previously implicated as TGF-β1 mediator, do not correlate with inhibition of CFTR mRNA (Figure 4, Figure 5, Figure 6 and Figure 7). Despite increasing miR-145 expression, TGF-β1 did not inhibit the CFTR mRNA in control HBE cells (Figure 4C). Although TGF-β1 inhibited CFTR mRNA and increased miR-145 expression in HBE cells from lungs with CF, COPD, or IPF, the miR-145 upregulation and CFTR mRNA inhibition did not correlate (Figure 4 and Figure 5). miR-145 expression is regulated by the canonical Smad2/3-mediated pathway [30]. Decreasing expression of endogenous miR-145 by Smad2, or Smad3 KD or by specific anti-miR-145 oligonucleotide did not increase or rescue CFTR mRNA expression (Figure 6 and Figure 7). Previously, sequencing of the total cellular miRNA population was thought to accurately reflect the active, i.e., RISC-associated, miRNA pool and was used as a surrogate of the inhibitory regulation of target genes by miRNAs. This view was recently challenged by data showing that only a small fraction, less than 10%, of miRNA is associated with RISC [72]. These and other data lead to the current model that the inhibitory potential of a miRNA is predicted better from its RISC association rather than its total cellular level [26]. The following evidence in our study supports this model. Despite increasing the total cellular expression of the predicted CFTR inhibitor miR-154 in F508del-CFBE cells, TGF-β1 did not recruit it to RISC (Figure 9F and Appendix A). By contrast, TGF-β1 increased the active pool of validated CFTR inhibitors, miR-143 and miR-145 by recruiting them to RISC without affecting their total cellular level in the CFBE41o- cell models (Figure 8 and Figure 9, and Appendix A). 

We do not know the specific mechanism(s) of how the three lung diseases enable miRNAs to destabilize CFTR mRNA. Overexpression of highly complementary mRNA targets facilitates the RISC recruitment of specific miRNAs targeting the genes [26]. Gene expression changes during lung disease. However, expression of CFTR mRNA was similar at baseline in the three lung diseases and controls (Figure 4C). Thus, the RISC recruitment of miR-143 and miR-145 is not facilitated directly by CFTR mRNA. The sequence complementarity with the target mRNA spans only a fraction of the miRNA sequence and each miRNA has hundreds of potential targets. Thus, altering abundance of the target mRNAs may also dilute the miRNA inhibitory effect [73]. It is conceivable that decreased expression of some transcripts during chronic lung disease may expose the CFTR mRNA to the destabilizing effect of miR-143, miR-145 or other miRNAs. 

In addition to the seed complementarity and thermodynamic stability governing the miRNA-mRNA interactions, other miRNA- and/or mRNA-specific factors fine-tune the process [25]. Recent evidence shows that the relative contribution and temporal characteristics of mRNA destabilization by miRNAs can be modulated by stress-induced signaling, environmental cues, epigenetic factors, developmental stage, mRNA methylation, or altered expression of protein adaptors regulating miRNA biogenesis, editing, or interactions with mRNAs [25,74,75,76]. Conditions affecting the RISC availability, its sub-cellular localization, and stoichiometry with the mRNA target sites, and the competition between miRNAs for RISC may have a major impact on the miRNA-mediated inhibitory gene regulation [77]. Our small RNA-seq data suggest that the F508del mutation may affect the competition between miRNAs for RISC because it globally modifies the miRNA expression in response to TGF-β1 (Figure 8, Table 2 and Appendix A). The data also show that TGF-β1 changes expression of miRNAs predicted to target CFTR 3’UTR and the F508del mutation also modulates their expression in response to TGF-β1 (Figure 8 and Appendix A). Whether TGF-β1 recruits these miRNAs to destabilize CFTR mRNA in a disease-specific manner will have to be determined by future studies. 

Our study did not examine how modifications of the miRNA target sites in the CFTR gene influence availability of CFTR mRNA to miRNAs. It has been shown that mutations and SNPs in the 3’UTR or 5’UTR, also present in the CFTR gene, may change the gene sensitivity to miRNAs, create novel binding sites, or even tilt the balance from health to disease [78,79,80,81,82]. Whether such modifications increase sensitivity of CFTR mRNA to TGF-β1 during chronic lung disease will also have to be addressed by future studies. 

In summary, our data increase understanding of the role of TGF-β1 in CF and other forms of chronic lung disease and may help to develop future strategies to achieve the full benefit of the transformative CFTR therapies in patients homozygous for F508del.

## 4. Materials and Methods 

### 4.1. Tissues, Cells and Cell Culture

Primary differentiated HBE cells were received from the CF Research Center Epithelial Cell Core at the University of Pittsburgh, School of Medicine, Pittsburgh, PA, as previously described [22,44,83]. The core procures these cells from excess pathologic lung tissue from explanted human lungs following lung transplantation at the University of Pittsburgh Medical Center under written consent and an approval by the Institutional Review Board (IRB) protocol REN18110058/IRB970946, last approved on November 20, 2018. The F508del HBE cells were derived from lungs homozygous for the F508del mutation. The COPD and IPF HBE cells were derived from lungs with COPD and IPF, respectively. Control HBE cells were isolated from tissue provided by the Center for Organ Recovery and Education and were provided by the University of Pittsburgh CF Research Center Epithelial Cell Core using previously described methods approved as described above [83]. Subsequently, cells were cultured on transwell filters (0.33 cm^2^ in density of ~2 × 10^5^/cm^2^) and maintained in ALI for 6 weeks while differentiation media was changed basolaterally twice weekly [22,44]. Formalin-fixed, paraffin-embedded human bronchial tissues from lungs described above were processed for ISH and IHC. Immortalized human bronchial epithelial cells CFBE41o- stably expressing WT- or F508del-CFTR were cultured in ALI as previously described [6,44]. HEK cells were cultured as previously described [41]. 

### 4.2. Antibodies and Reagents

The following primary mouse monoclonal antibodies were used, anti-CFTR (596) from Cystic Fibrosis Foundation Therapeutics (CFFT), Chapel Hill, NC [44]; anti-ezrin antibody 610603 from BD Biosciences (San Jose, CA, USA) anti-digoxigenin (DIG)-alkaline phosphatase antibody from Roche Applied Sciences (Indianapolis, IN, USA) and anti-Ago2 RN003M from Medical and Biological Laboratories (MBL) (Nagoya, Japan). The primary rabbit polyclonal antibodies, anti-TβR-I and anti-TβR-II (AP08190PU-N and AP54233PU-N, respectively) were from Acris (San Diego, CA, USA), anti-Smad2/3 5678 from Cell Signaling Technology, Inc. (Danvers, MA, USA). The horseradish peroxidase-conjugated goat anti-mouse and goat anti-rabbit (BioRad Laboratories, Hercules, CA, USA) were used. All antibodies were used at the concentrations recommended by the manufacturer and previously validated.

Human TGF-β1 (Sigma-Aldrich, St. Louis, MO, USA), or vehicle control (4 mM HCl + 1 mg/mL BSA) was used at a previously established clinically relevant concentration (15 ng/mL) based on measurements in serum and the blood of CF patients (2.8 to 35 ng/mL) [22,62,63,84]. Pierce^TM^ Protease Inhibitor Mini tablets EDTA-Free (Thermo Scientific, Rockford, IL, USA) was used in lysis buffer in the RIP assay kit (MBL). Corrector VX-809 was from Selleck Chemicals (Houston, TX, USA), and the correctors C18 (VRT-534) and CFFT-002 were from CFFT. All correctors were used at 5 μM concentration to increase delivery of F508del-CFTR to the apical membrane in HBE cells. Correctors, or vehicle-control DMSO were added daily to basolateral medium of F508del HBE cells for 48 h [22]. The final concentration of DMSO was <0.1%. The CFTR_inh_-172 (5 μM) was from Selleck Chemicals. Amiloride (50 μM), forskolin (20 μM), and IBMX (1 mM) were from Sigma-Aldrich. 

### 4.3. Western Blotting

Western blotting was performed using the Western Lightning™ Plus-ECL detection system (Perkin Elmer Inc., Waltham, MA, USA) followed by chemiluminescence. Protein abundance was quantified by densitometry using exposures within the linear dynamic range of the film [44,85]. 

### 4.4. Short Circuit Recordings

Isc were measured under asymmetrical Cl^−^ conditions, as previously described [22]. In brief, 6-week ALI cultures of HBE cells were mounted in Ussing-type chambers (Physiological Instruments, San Diego, CA, USA). The composition of the bathing Ringer’s solutions was as follows, apical: 115 mM NaCl, 25 mM NaHCO_3_, 5.0 mM KCl, 10 mM HEPES, 1.0 mM MgCl_2_, 1.5 mM CaCl_2_ and 5.0 mM glucose; and basolateral: 114 mM sodium gluconate, 25 mM NaHCO_3_, 5.0 mM KCl, 10 mM HEPES, 1.0 mM MgCl_2_, 1.5 mM CaCl_2_ and 5.0 mM glucose. Chambers were constantly gassed with a mixture of 95% O_2_ and 5% CO_2_ at 37 °C, which maintained the pH at 7.4. Following a 5 min equilibration period, the baseline Isc was recorded. Amiloride (50 µM) was added to the apical bath solution to inhibit Na^+^ absorption through ENaC. Subsequently, Isc was stimulated with the cAMP agonist forskolin (20 µM) together with IBMX (1 mM) to prevent cAMP hydrolysis, added to the apical and basolateral bath solutions, followed by VX-770 (5 μM) added to the apical bath to increase the open probability of the CFTR channels (only in F508del-CFTR expressing cells), and subsequently followed by CFTR_inh_-172 (5 µM) added to the apical bath solution to inhibit CFTR-mediated Isc. Data are expressed as the forskolin/IBMX/VX-770-stimulated Isc (F508del-CFTR) and the forskolin/IBMX-stimulated Isc (WT-CFTR), calculated by subtracting the peak-stimulated Isc from the baseline Isc after amiloride treatment.

### 4.5. ASL Volume Measurements

Measurements of ASL height were carried out by meniscus scanning using CannonScan8800 [50]. Scanned menisci data were transformed to ASL volume using an ImageJ-based custom algorithm [86].

### 4.6. CFTR Gene Promoter Reporter Assay

To study the TGF-β1 effect on the CFTR promoter activity, we used the firefly luciferase construct pCFTR-pLuc containing approximately a 1 kb fragment of the 5’UTR of the human CFTR gene (Catalog #LR1020, Panomics Inc., Fremont, CA, USA). The assay was performed in HEK cells, according to the manufacturer’s instructions and as previously described [51,52]. Briefly, HEK cells were seeded on 6-well tissue culture plates (1 × 10^5^/well) and each transfected with the pCFTR-pLuc construct (1 µg/well) or the TransLucent control vector (Panomics), using FuGENE6 (Roche Applied Science). At 24 h post-transfection, cells were treated with forskolin (10 µM), TGF-β1 or vehicle in medium without fetal bovine serum (FBS) for 6 h. Subsequently, cells were lysed using luciferase assay lysis buffer (Promega) and firefly/Renilla luciferase activities were measured using the Dual-Luciferase Reporter Assay (Promega), according to the manufacturer’s protocol.

### 4.7. mRNA Degradation Measurements

CFTR mRNA decay was measured in 6-week ALI cultures of F508del HBE cells in the presence of the transcriptional inhibitor ActD by the method described previously [51,52]. The stock solution of ActD, dissolved in DMSO, was used at a concentration of 5 µg/mL in cell culture medium with a final DMSO concentration < 0.1%. Total RNA was isolated 4, 8, and 24 h later, and CFTR mRNA was measured by qRT-PCR, using the relative quantitation method, as described in Section 4.11 below. 18S was used as a reference gene because there was no degradation of its transcripts even after 24 h treatment with ActD. Data were first plotted as % mRNA remaining, normalized to 18S versus time, with the first timepoint (4 h) set to 100%. Data were normalized to the 4 h timepoint to allow time for ActD to act, eliminate any potential DMSO effects, avoid initial feedback-driven transcriptional enhancement, and allow for initial poly-A tail deadenylation that precedes the first-order degradation kinetics. CFTR mRNA half-lives were calculated from the exponential decay model, based on the trendline equation C/C_0_ = e^−kdt^ (where C and C_0_ are mRNA amounts at the time t and t_0_, respectively, and k_d_ is the mRNA decay constant). 

### 4.8. In Situ Hybridization and lna™ Probes for miR-145

ISH was performed by methods described previously [87,88]. All steps were carried out using RNase-free reagents and equipment. Formalin-fixed, paraffin-embedded 5 µm tissue sections were mounted onto glass slides and incubated at 55 °C overnight. Slides were placed in xylene and then hydrated through ethanol solutions to phosphate-buffered saline (PBS). From here, the Exiqon protocol for LNA^TM^ miR-145 probe for ISH was followed (Exiqon, Woburn, MA, USA). In brief, RNA was exposed during a 15 min proteinase K incubation, followed by a PBS rinse and sequential step dehydration through increasing gradients of ethanol solutions. Hybridization buffer with LNA^TM^ miR-145 probe was prepared according to company directions at 1:100 dilution. Dehydrated slides were incubated for 1 h at 55 °C and covered with parafilm to prevent drying. Parafilm was gently removed and slides were placed through decreasing concentrations of warm saline–sodium citrate solutions. Slides were then permeabilized in Tris-buffered Saline with 1% Tween-20 (TBST). After permeabilization, sections were isolated with a hydrophilic barrier using a peroxidase, anti-peroxidase (PAP) pen (Abcam Inc., Cambridge, MA, USA) and incubated with blocking buffer (Roche Applied Sciences, Indianapolis, IN, USA) at room temperature (RT) for 1 h, followed by incubation with anti-DIG-alkaline phosphatase antibody (Roche) 1:1000 in blocking buffer overnight at 4 °C in a humidifying chamber filled with TBST. The next day, slides were washed twice with TBST containing 2 mM levamisole and incubated at RT for 15 min in 2 mM levamisole in AP Buffer (5 mM NaCl, 50 mM MgCl_2_, 10% Tween-20, 100 mM Tris-HCl) to remove any endogenous phosphatases. Slides were then incubated with BM Purple AP substrate precipitating solution (Roche) at RT for 24 h in the dark for color development. Once the color change was deemed sufficient, slides were rinsed in water and dehydrated through an increasing gradient of ethanol solutions and followed by xylene treatment and mounted in the mounting medium before coverslipping. Control slides were stained with nuclear fast red (Vector Laboratories, Burlingame, CA, USA) at RT for 1 min.

### 4.9. Immunohistochemistry

Formalin-fixed, paraffin-embedded tissue sections were stained on the Ventana BenchMark Ultra automated staining platform (Ventana Medical Systems Inc., Tucson, AZ) using the following protocols. Slides were pretreated with ULTRA cell conditioning solution CC1 (Ventana) for 64 min and stained using rabbit polyclonal primary antibodies against the N-terminus of TβR-I, or TβR-II (AP08190PU-N, and AP54233PU-N, respectively, at 1:100 dilution; Acris, San Diego, CA, USA). OptiView 3,3’-diaminobenzidine (DAB) IHC detection kit with OptiView amplification indirect, biotin-free multimer amplification system (Ventana) was used for detection of primary antibodies. All slides were counterstained with hematoxylin and routinely dehydrated, cleared, and coverslipped in resinous mounting media.

### 4.10. RNA Isolation

Total RNA was isolated from ALI cultures of HBE cells using RNeasy mini kit (Qiagen, Valencia, CA, USA) and Quick-RNA Miniprep Kit (Zymo Research, R1054) according to the manufacturer’s instructions, with additional on-column DNase treatment with the RNase-free DNase Set (Qiagen/Zymo) to remove contaminating genomic DNA for downstream applications, as described previously [22]. RNA quantity and quality were measured by NanoDrop. 

### 4.11. Real-Time Quantitative Reverse-Transcription PCR 

Real-time reactions were run in triplicate with each reaction emanating from a starting sample amount of 20 ng total RNA before reverse transcription to cDNA. Superscript II reverse transcriptase (Invitrogen, Grand Island, NY) was used to generate cDNA from total RNA. qRT-PCR was performed using ABsolute™ Blue QPCR SYBR^®^ Green ROX Mix (Thermo Scientific, Walthman, MA) and ABI PRISM^®^ 7300 Sequence Detection System (Applied Biosystems, Foster City, CA), according to the manufacturer’s instructions. The primer sequences for CFTR were from the Harvard Medical School Primer Bank (ID#09421312c2; CFTR-213 forward: 5’- TGCCCTTCGGCGATGTTTTT-3’ and CFTR-339 reverse: 5’-GTTATCCGGGTCATAGGAAGCTA-3’) [22]. The primer sequences for GAPDH RNA were forward:5’-TGACAACTTTGGTATCGTGGAAGG-3’ and reverse:5’-AGGGATGATGTTCTGGAGAGCC-3’ [22]. Primer sequences for 18S RNA were forward:5’-GGACATCTAAGGGCATCACAG-3’ and reverse:5’-GAGACTCTGGCATGCTAACTAG-3’. Fluorescence emission was detected for each PCR cycle and the Ct values and the average Ct of the triplicate reactions were determined for CFTR and GAPDH or 18S. The Ct value was defined as the actual PCR cycle when the fluorescence signal increased above the threshold, the ∆Ct was determined for each sample by subtracting the Ct for the reference gene (GAPDH or 18S) from the Ct for CFTR, and the average ∆Ct of the triplicate samples was determined. The ∆∆C*t* was calculated by subtracting the ∆Ct for the vehicle-treated cells from the ∆Ct for the TGF-β1 treated cells. FC values were determined according to the formula: FC = 2^−∆∆C*t*^. Log2 FC was calculated by converting FC value in log base 2 in Microsoft Excel.

### 4.12. miRNA qRT-PCR

miRNA levels were quantified with qRT-PCR using the relative quantitation method. Validated reverse transcription primers, qPCR primers, and qPCR hydrolysis probes were supplied together in each assay set. First, 100 ng total RNA was reverse transcribed to generate cDNA with stem-loop primers specific for miR-145-5p, miR-143-5p, and miR-154-5p (Thermo Fisher Scientific Cat# 4427975, assay ID #002278, #002146, and #000477, respectively) or U6 sn RNA (Thermo Fisher Scientific, assay #001973) using the TaqMan MicroRNA Reverse Transcription Kit (Applied Biosystems, cat# 4366596) according to manufacturer’s instructions. Next, qPCR was performed using 1.33 μL cDNA of miR-145, miR-143, miR-154, or U6-specific qPCR primer sets containing miRNA-specific forward and reverse primers and hydrolysis probes and TaqMan Universal Master Mix II (Applied Biosystems, PN 4440040) in 20 μL reactions in triplicate. qPCR samples were run for 45 cycles on an Applied Biosystems 7300 thermocycler. qPCR curve baselines were set manually per plate and thresholds were standardized for all qPCR runs at a point at which amplification was logarithmic for all samples. To quantify relative miRNA levels between samples, U6 Cts were subtracted from miRNA Cts to generate ΔCts. Finally, FC in miRNA levels between samples were determined using the ΔΔCt method.

### 4.13. miR-145 Inhibitor and Analog

Antisense oligonucleotides form duplexes and sequester the respective miRNA, leading to inhibition of the target miRNAs [53,89,90], hence, the name “miRNA inhibitors”. The cellular uptake of miRNA inhibitors is an active process that does not require carriers or transfection reagents. Modifications of the nucleic acids as 2’-*O*-methyl, 2’-*O*-methoxyethyl and LNAs increase cellular stability. ALI cultures of F508del HBE cells were treated with 50 nM LNA-antisense probe (miRCURY LNA™ Power Inhibitor hsa-miR-145-5p; Exiqon) or control (miRCURY LNA™ Inhibitor Control), added to the basolateral medium every other day for 8 days. CFTR correctors were added daily for 48 h, and TGF-β1 or vehicle control was added for 24 h before experiments. 

ALI cultures of HBE cells were treated with 5 nM miRCURY LNA™ Power Mimic (has-miR-145-5p; Exiqon), referred to as miR-145 analog or analog control (miRCURY LNA™ Power Negative control), and added to the basolateral medium with HiPerFect Transfection reagent (Qiagen) for 48–72 h. Following the above treatments with either miRNA inhibitors or analog, cells were lysed and mRNA and miRNA levels were quantified by qRT-PCR.

### 4.14. RNA-Mediated Interference

F508del HBE cells were plated on tissue culture plates and after reaching 50% confluence, cells were incubated with the optimized transduction mixture containing the shRNAmir targeting the human Smad2 gene (shSmad2; V2LHS_261805, V2LHS_340703, or V2LHS_213986), Smad3 gene (shSmad3; V2LHS_215032, V2LHS_359687, or V2LHS_359688), or the shRNAmir not directed against any human gene known (shControl; RHS4348) in the lentiviral vector pGIPZ with TURBO-GFP reporter and the SV40-T antigen (Open Biosystems, Huntsville, AL). At 72 h after transduction, puromycin was added daily to the cell culture medium to select the transduced cells. Puromycin-selected cells were cultured until 90% confluence and transferred to collagen-coated transwell filters (1.12 cm^2^ at density 7 × 10^5^/cm^2^) and cultured in ALI for 6 weeks to form differentiated monolayers in medium supplemented with puromycin. The V2LHS_261805 shSmad2 depleted Smad2 protein in differentiated F508del HBE cells by at least 80% and was used in subsequent experiments. By contrast, neither of the three shSmad3, individually or pooled together, depleted Smad3 by more than 46%. 

### 4.15. Small RNA Library Preparation, Data Analysis, and miRNA Profiling

WT- or F508del-CFBE cells cultured in ALI were treated for 24 h with TGF-β1 or vehicle control in six separate experiments. Total RNA was isolated using the Quick-RNA Miniprep Kit (Zymo Research, R1054), according to the manufacturer’s instructions, with additional on-column DNase treatment with the RNase-free DNase Set (Zymo) to remove contaminating genomic DNA for downstream applications, as described previously [22]. The starting material, 100 ng of total RNA was used with the QIAseq^TM^ miRNA Library Kit (Qiagen, Maryland, MD, USA), following the manufacturer’s instruction. The RNA quantity and quality were assessed by NanoDrop, Qubit 2.0 Fluorometer, and Agilent Bioanalyzer Tapestation 2200. In short, 3’ and 5’ adapters were ligated to total input RNA. Reverse transcription, followed by PCR, was used to create cDNA constructs. This process selectively enriches those fragments that have adapter molecules on both ends. PCR was performed with primers that anneal to the ends of the adapters. Quality was examined using Agilent Bioanalyzer Tapestation 2200 and Qubit 2.0 Fluorometer with miRNA library pre-sequencing QC protocol mentioned in the QIAseq^TM^ miRNA Library Kit. The cDNA libraries were pooled at a final concentration of 2.5 pM. Cluster generation and 75 base-pair (bp) single-read single-indexed sequencing was performed on Illumina NextSeq 500’s. microRNA sequencing data were analyzed using the GeneGlobe Data Analysis bioinformatics pipeline from Qiagen [91]. The raw QIAseq miRNA sequencing files were uploaded to GeneGlobe Data Analysis Center (Qiagen) for quality control, alignment to the human reference genome (GRCh38), and expression quantification. Briefly, 3’adapter and low-quality bases were trimmed off from reads using Cutadapt. Reads with less than 16 bp insert sequences or with less than 10 bp unique molecular identifier (UMI) sequences were discarded. The remaining reads were collapsed to UMI counts and aligned to miRBase (release v21) mature and hairpin databases sequentially using Bowtie v1.2. All reads assigned to a particular miRNA or piRNA ID were counted and the associated UMIs were clustered to count unique molecules. The primary analysis output file was downloaded from GeneGlobe. The edgeR package was used for the differential analysis based on UMI counts for miRNA. miRNAs were filtered from the analysis if their counts per million (as computed by edgeR’s cpm function) were not greater than 2 in at least half of the samples. The trimmed mean of *M*-values normalization method implemented in edgeR was used for normalization. miRNAs predicted to target the human CFTR 3’UTR were collected from TargetScan Human release 7.2, miRmap, Diana-TarBase v.8, and miRBase bioinformatics tools [54,55,57,59].

### 4.16. RISC IP Assay

The RIP assay kit from MBL was used to examine how TGF-β1 affects the interaction of miRNAs with the endogenous Ago2 of RISC, according to manufacturer’s instructions and as previously described [92]. The IP antibody complexes were prepared by conjugating 15 μg of RIP-certified anti-EIF2C2 (Ago2) mouse monoclonal antibody (MBL, RN003M) with 30 μL of 50% Pierce^TM^ Protein G sepharose bead slurry (Thermo Scientific, Rockford, IL) overnight at 4 °C. Mouse IgG was used as a negative control (Dako, Glostrup, Denmark). CFBE41o- cells stably expressing WT-CFTR or F508del-CFTR, treated for 24 h with TGF-β1 or vehicle control were lysed and the supernatants precleared with unconjugated Protein G sepharose beads to reduce nonspecific adsorption. Subsequently, precleared supernatants were incubated with the Ago2-Protein G complexes for 3 h at 4 °C. Ago2 IP was confirmed by WB using the anti-Ago2 primary antibody and horseradish peroxidase-conjugated goat anti-mouse secondary antibody. Small and large RNA was isolated from the IP complexes through a separation method as per the manufacturer’s protocol and miRNA levels were measured by qRT-PCR, as described above. To quantify relative miRNA levels between samples, vehicle control miRNA Cts were subtracted from the TGF-β1-treated miRNA Cts to generate ΔCts. Finally, fold changes in miRNA levels between samples were determined using the equation: FC = 2^−ΔC*t*^.

### 4.17. Data and Code Availability

The small RNA-seq data have been deposited in NCBI’s Gene Expression Omnibus (GEO) and are accessible through GEO accession number GSE128765.

### 4.18. Data Analysis and Statistics

Statistical analysis of data was performed using GraphPad Prism version 8.0 for Mac OS X (GraphPad Software Inc., San Diego, CA, USA). Data are expressed as mean ± S.E.M. The means were compared by a two-tailed *t*-test. The half-lives were calculated using the one-phase exponential decay model, with plateau, and span parameters constrained to zero [6]. For analyzing the TGF-β1-induced change in miR-145 levels, or CFTR mRNA expression, a two-sided Fisher’s exact test was used to statistically compare groups. For analyzing the correlation between the TGF-β1-induced change in miR-145 level and CFTR mRNA expression, Spearman correlation analysis was performed and the correlation coefficient (r) was calculated. For the small RNA-seq data (GEO accession number GSE128765), the means were compared by two-sided *t*-test, and considered significant when −log10 > 1.3, corresponding to *p* < 0.05. *p* < 0.05 was considered significant where * *p* < 0.05; ** *p* < 0.01; *** *p* < 0.001; **** *p* < 0.0001. 

## Figures and Tables

**Figure 1 ijms-20-04933-f001:**
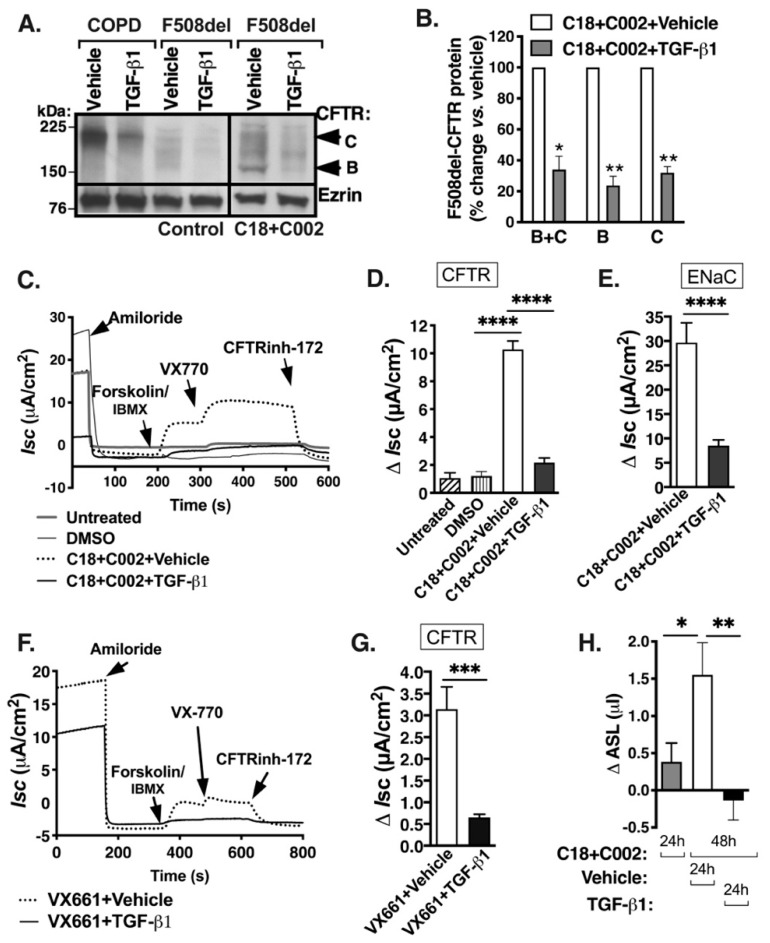
TGF-β1 inhibits corrector rescue of the F508del-Cystic Fibrosis Transmembrane Conductance Regulator (CFTR) protein, channel function and airway surface liquid (ASL) volume in F508del HBE cells. Representative WB images (**A**) and summary of data (**B**) showing that the combined use of corrector C18 and C002 increased the abundance of the partially glycosylated CFTR band B and the fully glycosylated band C in F508del human bronchial epithelial (HBE) cells, compared to control, while TGF-β1 prevented the corrector-rescue of the CFTR band B and C. Six-week ALI cultures of F508del HBE cells were treated with C002 and C18 (5 μM each) for 48 h, and with TGF-β1 (15 ng/mL) or vehicle control (HCl/BSA) for 24 h before cell lysis. Non-cystic fibrosis (CF) HBE cells from chronic obstructive pulmonary disease (COPD) lungs were used as a positive control for CFTR protein abundance. Ezrin was used as a loading control to normalize CFTR expression. Representative Ussing chamber recordings (**C**) and summary of data (**D**,**E**) showing that correctors C18/C002 and potentiator VX-770 rescued the CFTR-mediated Isc, compared to controls with no treatment or dimethyl sulfoxide (DMSO) control used as correctors’ vehicle. TGF-β1 blocked rescue of the CFTR-mediated Isc, and inhibited the ENaC-mediated Isc. Amiloride (50 µM) was added to the apical bath solution to inhibit Na^+^ absorption through ENaC. Forskolin (20 µM) and IBMX (1 mM) were added to increase cAMP level, followed by VX-770 (5 µM) to increase the CFTR channel open probability, and CFTR_inh_-172 (5 µM) was added to inhibit CFTR channel function. The CFTR ΔIsc was calculated by subtracting the Isc after amiloride treatment from the peak forskolin/IBMX/VX770-stimulated Isc. The ENaC ΔIsc was calculated by subtracting the Isc after amiloride treatment from the baseline Isc before any treatment. Representative recordings (**F**) and summary of data (**G**) demonstrating that TGF-β1 blocked the CFTR-mediated Isc rescued by corrector VX-661 in F508del HBE cells (**H**). Summary of ASL volume measurements. ASL volume in ALI cultures of F508del HBE cells was measured by scanning the apical meniscus before treatment. Cells were scanned, treated with correctors C18/C002 for 24 h, and scanned again. Subsequently, cells were treated with TGF-β1 or vehicle in the presence of correctors C18/C002 and scanned again 24 h later. ^Δ^ ASL represents the change from baseline in ASL volume at 24 h after corrector rescue or at 48 h after corrector rescue with TGF-β1 of vehicle treatment. Corrector C18/C002 increased the ASL volume while TGF-β1 blocked the rescue. Experiments were repeated four times in HBE cells from two donors/group (**A**,**B**). Eight monolayers/group from two lung donors (**C**, **D** and **G**), six monolayers/group from two lung donors (**E**,**F**), and 11 monolayers/group from two lung donors (**H**) were used. Error bars, standard error of the mean (S.E.M.). * *p* < 0.05; ** *p* < 0.01; *** *p* < 0.001; **** *p* < 0.0001.

**Figure 2 ijms-20-04933-f002:**
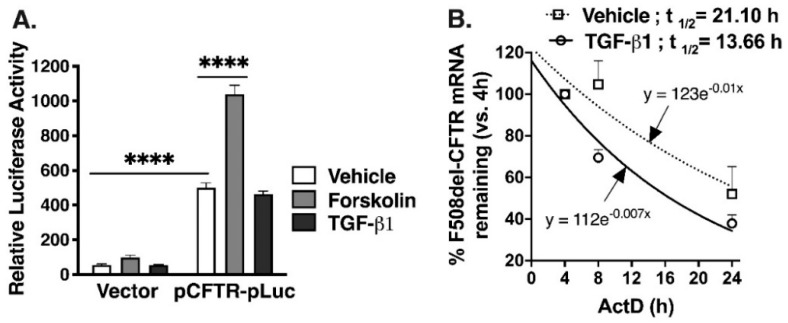
TGF-β1 did not affect CFTR promoter activity but reduced the stability of CFTR mRNA. (**A**). Summary of the CFTR gene promoter reporter assays showing that luciferase activity was increased by expression of the vector containing 1 kb promoter pCFTR-pLuc, compared to vector control. Forskolin increased the luciferase activity of pCFTR-pLuc and TGF-β1 had no effect. Human embryonic kidney (HEK) cells were transfected with pCFTR-pLuc or vector control and 24 h later cells were treated with forskolin (10 µM), TGF-β1 (15 ng/mL), or vehicle control for 6 h. (**B**). The CFTR mRNA half-life measurements showing that TGF-β1 decreased CFTR mRNA stability in F508del HBE cells. The transcriptional inhibitor ActD (5 μg/mL) and either TGF-β1 or vehicle control were added to F508del HBE cells for 4, 8 or 24 h. Total RNA was isolated and the CFTR mRNA level at each time point was calculated by real-time quantitative reverse transcriptase PCR (qRT-PCR), using the relative quantitation method, with 18S as a reference gene. The amount of CFTR mRNA at 4 h was set to 100% (*t* = 0) and mRNA half-lives were calculated from the exponential decay model, based on trend line equation C/C_0_ = e^−kdt^ (where C and C_0_ are mRNA amounts at the time t and t_0_, respectively, and k_d_ is the mRNA decay constant). The resulting curve equations were y_(vehicle)_ = 123^−0.01x^ and y_(TGF-β1)_ = 112^−0.007x^. The calculated half-life of CFTR mRNA was 21.1 h and 13.7 h for the vehicle and TGF-β1-treated cells, respectively. *N* = 9–12 /group from 3–4 different HEK cell cultures (**A**) and *N* = 3 in triplicates in F508del HBE cells from three different donors (**B**). Error bars, S.E.M. **** *p* < 0.0001.

**Figure 3 ijms-20-04933-f003:**
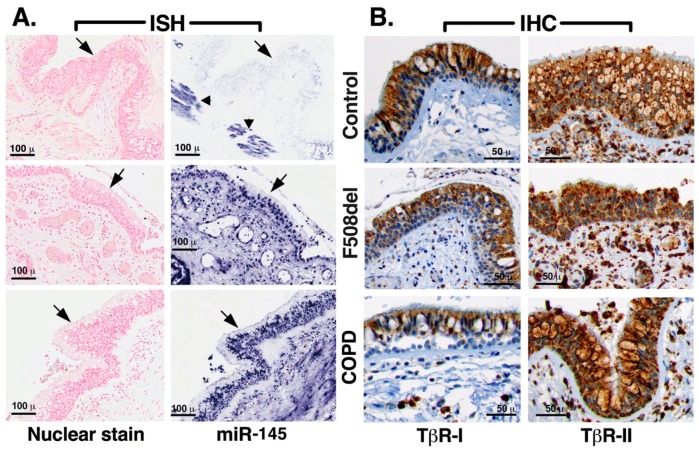
miRNA-145 expression was increased in native human bronchial epithelial samples from lung disease patients. (**A**). ISH images showing increased expression of miR-145 (arrows) in sections of bronchial epithelia from lungs homozygous for F508del (*n* = 3) and COPD (*n* = 2), compared to epithelia from lungs without chronic disease (control; *n* = 3). Previously reported high expression of miRNA-145 in the airway SMCs (arrowheads) was also noted. miR-145 was detected by locked nucleic acids (LNAs)™ miR-145 probe for ISH (Exiqon) and red nuclear stain was used as a counter stain. (**B**). IHC images showing similar expression of TβR-I and TβR-II in the CF and non-CF bronchial epithelia. TβR-I and TβR-II were detected with antibody AP08190PU-N and AP54233PU-N, respectively. Triplicate samples from each bronchial tissue per condition were used.

**Figure 4 ijms-20-04933-f004:**
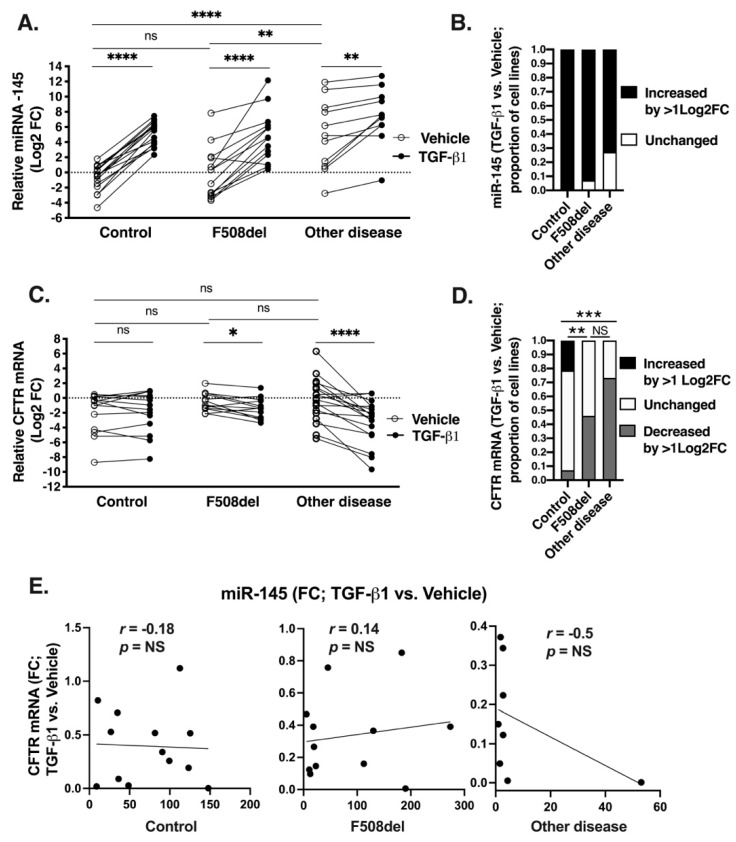
TGF-β1 upregulation of miR-145 expression did not correlate with inhibition of CFTR mRNA in HBE cells. (**A**). qRT-PCR experiments showing relative expression of miR-145 after treatment with vehicle or TGF-β1 added for 24 h to the basolateral medium of ALI cultures of HBE cells originating from control HBE cells (*n* = 16), F508del (*n* = 14), or other disease (*n* = 11). miR-145 expression was normalized to U6 small nuclear (sn)RNA and expressed as Log2 fold change (FC) relative to the mean of the first three examined control cell lines, treated with vehicle. (**B**). Proportions of the HBE cell lines, shown in A, with increased miR-145 expression by > 1 Log2 FC and unchanged (< 1 Log2 FC). (**C**). qRT-PCR experiments showing the relative expression of CFTR mRNA after treatment with vehicle or TGF-β1, as described above, in control (*n* = 14), F508del (*n* = 13), or other disease (*n* = 18). HBE cell lines CFTR mRNA was normalized to glyceraldehyde-3-phosphate dehydrogenase (GAPDH) mRNA and expressed as Log2 FC relative to the mean of the first three examined control cell lines, treated with vehicle. (**D**). Proportions of the HBE cell lines, shown in C, with increased CFTR mRNA expression by > 1 Log2 FC, decreased by > 1 Log2 FC, and unchanged. All experiments in all cell lines were performed twice in triplicates. Error bars, S.E.M. * *p* < 0.05; ** *p* < 0.01; *** *p* < 0.001; **** *p* < 0.0001, TGF-β1 versus vehicle control (**A**,**C**), proportions of cell lines with miR-145 expression > 1 Log2 FC versus unchanged (**B**), and proportions of cell lines with CFTR mRNA expression > 1 log2 FC versus unchanged and decreased (**D**). (**E**). Spearman’s correlation analysis and correlation coefficient (r) of paired miR-145 and CFTR mRNA FC after TGF-β1 treatment versus vehicle in HBE cells used in (**A**–**D**). There was no significant correlation between the changes in miR-145 and CFTR mRNA expression in the three categories of HBE cells. FC outliers were considered if > upper bound (Q_3_+(1.5*IQR)) or < lower bound (Q_1_−(1.5*IQR)) and were removed. Q_3_, the third quartile; Q_1_, the first quartile; IQR, interquartile range.

**Figure 5 ijms-20-04933-f005:**
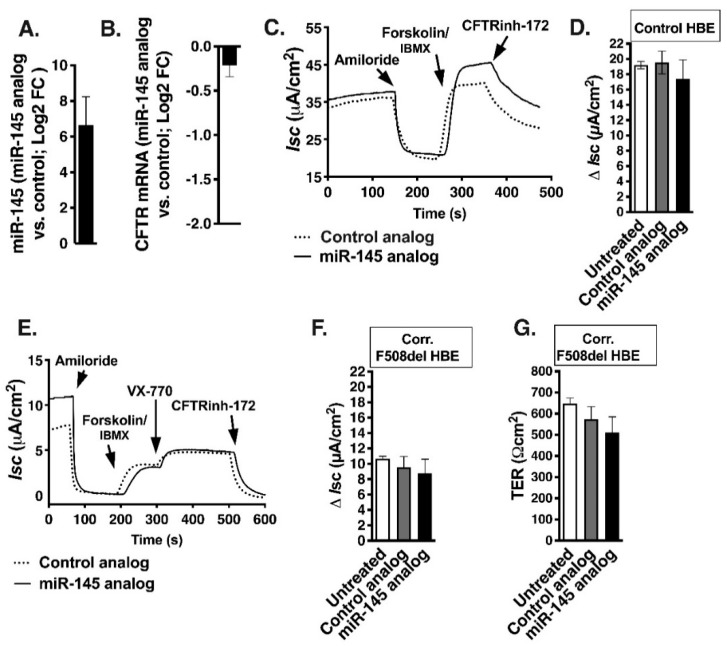
Increasing miR-145 levels by synthetic miR-145 analog did not change CFTR mRNA expression and CFTR-mediated *I*sc in control HBE cells or F508del HBE cells rescued by correctors C18/C002 (Corr. F508del HBE). Cells were treated with miR-145 analog or control analog for 48–72 h. (**A**). qRT-PCR experiments demonstrating that miR-145 analog increased the total cellular miR-145 level compared to control analog. Expression of miR-145 after miR-145 analog treatment was normalized to U6 snRNA and expressed as Log2 FC versus control analog. (**B**). qRT-PCR experiments demonstrating that increasing the total cellular level of miR-145 did not inhibit CFTR mRNA. Expression of CFTR mRNA after miR-145 analog treatment was normalized to GAPDH mRNA and expressed as Log2 FC versus control analog. Representative Ussing chamber recordings (**C**) and summary of data (**D**) demonstrating that increasing the total cellular miR-145 level did not affect the CFTR-mediated Isc. Representative recordings (**E**) and summary of data (**F**) showing that increasing the total cellular miR-145 level did not affect the corrector-rescued F508del-CFTR-mediated Isc. (**G**). The experimental conditions did not change the transepithelial resistance (TER) across the cell monolayers, measured after equilibration of cells in Ussing chamber and before the addition of drugs. Ten monolayers/group from four lung donors (**A**,**B**), eight monolayers/group from one lung donor (**C**,**D**), and 6–8 monolayers/group from one lung donors (**E**–**G**) were used. Error bars, S.E.M.

**Figure 6 ijms-20-04933-f006:**
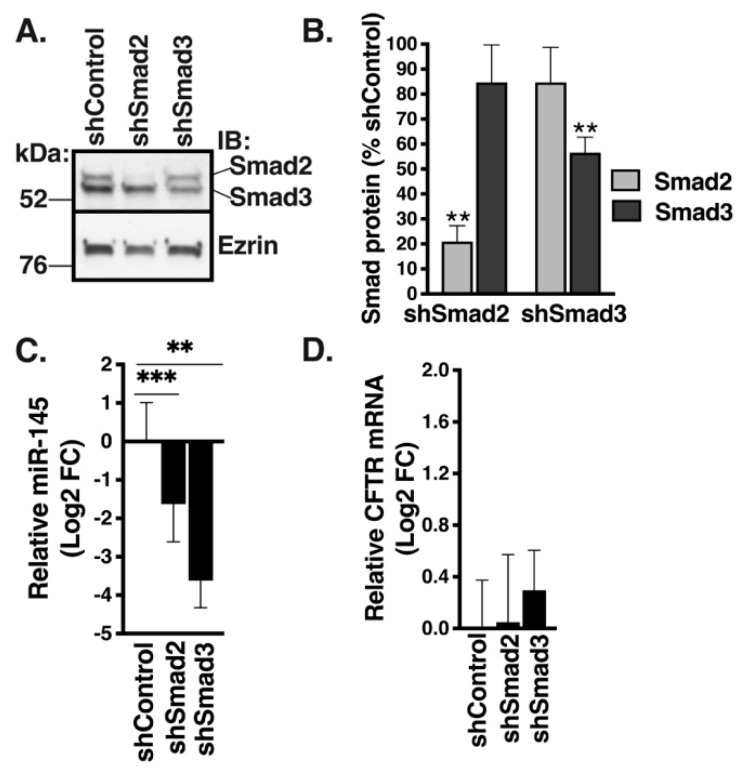
Inhibiting endogenous miR-145 expression did not change CFTR mRNA levels in F508del HBE cells. F508del HBE cells were transduced with shSmad2, shSmad3, or shControl and cells were cultured in ALI for 6 weeks. Representative WB images (**A**) and summary of data (**B**) showing that shSmad2 depleted Smad2 protein and had no significant effect on Smad3, while shSmad3 decreased Smad3 abundance without affecting Smad2. (**C**). qRT-PCR experiments demonstrating that knockdown (KD) of Smad2 or Smad3 significantly decreased miR-145 expression. Expression of miR-145 after KD of Smad2 or Smad3 was normalized to U6 snRNA and expressed as Log2 FC versus shControl. (**D**). qRT-PCR experiments demonstrating that neither Smad2 KD nor Smad3 KD affected CFTR mRNA expression. Expression of CFTR mRNA after KD of Smad2 or Smad3 was normalized to GAPDH mRNA and expressed as Log2 FC versus shControl. Inhibiting endogenous expression of miR-145 did not affect the CFTR mRNA expression. *N* = 5 from 2 F508del HBE cell donors/group. Error bars, S.E.M. ** *p* < 0.01; *** *p* < 0.001.

**Figure 7 ijms-20-04933-f007:**
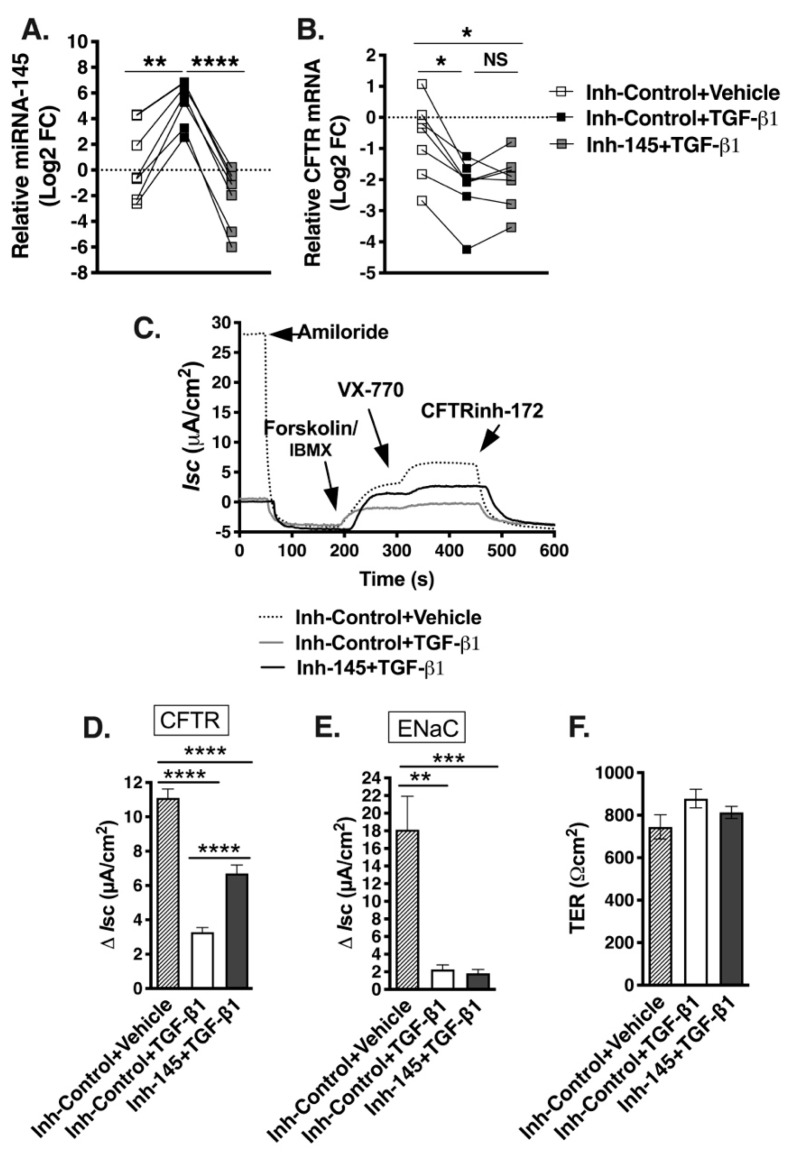
Anti-miR-145 oligonucleotide completely blocked TGF-β1 upregulation of miR-145 but only partially reversed repression of CFTR mRNA, allowing partial rescue of CFTR Isc in select F508del HBE cells. F508del HBE cells were treated with miRCURY LNA™ Power Inhibitor hsa-miR-145-5p (Inh-145) or miRCURY LNA™ Inhibitor control (Inh-Control) for 8 days, correctors C18 and C002 for 48 h, and with either TGF-β1 or vehicle for 24 h before the experiments. (**A**). qRT-PCR experiments demonstrating that Inh-145 blocked the TGF-β1-induced upregulation of miR-145 expression in all examined HBE cells. Expression of miR-145 was normalized to U6 snRNA and expressed as Log2 FC relative to Inh-Control and vehicle-treated cells. (**B**). qRT-PCR experiments demonstrating that Inh-145 partially blocked the TGF-β1 repression of F508del-CFTR mRNA in four out of seven examined HBE cell lines. Expression of CFTR mRNA was normalized to GAPDH mRNA and expressed as Log2 FC relative to Inh-Control and vehicle-treated cells. Representative Ussing chamber recordings (**C**) and summary of data (**D**) showing that Inh-145 partially blocked the TGF-β1 inhibition of corrector rescued CFTR-mediated Isc in those F508del HBE cells in which Inh-145 partially blocked repression of CFTR mRNA. (**E**). Inh-145 did not affect the TGF-β1 inhibition of ENaC-mediated Isc. (F). The experimental conditions did not change the trans-epithelial resistance (TER) across F508del HBE cell monolayers. *N* = 7 in triplicates in F508del HBE cells from seven different donors (**A**,**B**). Eight monolayers/group from two lung donors (**C**–**F**). Error bars, S.E.M. * *p* < 0.05; ** *p* < 0.01; *** *p* < 0.001; **** *p* < 0.0001.

**Figure 8 ijms-20-04933-f008:**
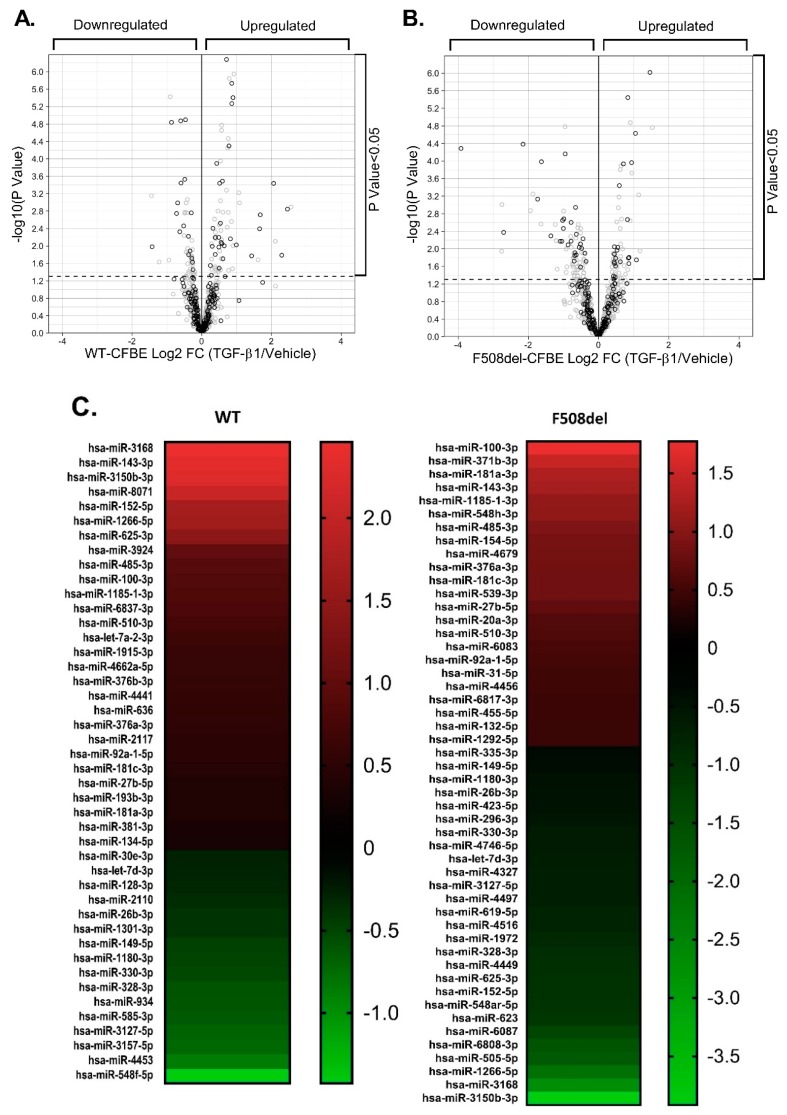
Small RNA-seq data showing effects of TGF-β1 on the miRNA landscape. Volcano plots of data from WT-CFBE (**A**) and F508del-CFBE (**B**) cells. TGF-β1 changed expression of many miRNAs predicted to target (black circles) and those not predicted to target (grey circles) the human CFTR 3’UTR. Cells were treated with TGF-β1 or vehicle for 24 h. The −log10 (*p* value) >1.3 corresponding to *p* < 0.05 was considered significant. Only shown in Appendix A, due to the limitation of volcano plot cut-off area, are hsa-miR-143-3p significantly upregulated in WT- and F508del-CFBE cells with −log10 (*p* value) = 53.78 and 8.42, respectively, hsa-miR-100-3p, and hsa-miR-181a-3p significantly upregulated in F508del CFBE with -log10 (*p* value) = 15.7 and 7.7, respectively. Upregulation of the miR-143-5p (referred to as miR-143) and miR-145-5p (referred to as miR-145) by TGF-β1 did not reach statistical significance in either WT- or F508del-CFBE cells. (**C**). Heatmap of miRNAs predicted to target human CFTR 3’UTR that were dysregulated by TGF-β1. These miRNAs are marked with black circles in A and B. Only miRNAs with −log10 (*p* value) > 1.6 (*p* < 0.025) are shown. Sequences of miRNAs were extracted from the miRBase database and were used in the miRmap database to search their predicted binding in CFTR 3’UTR. The change in miRNA expression after TGF-β1 treatment versus vehicle control is shown in the legend as Log2 FC. *N* = 6 in triplicates from six different cultures/group.

**Figure 9 ijms-20-04933-f009:**
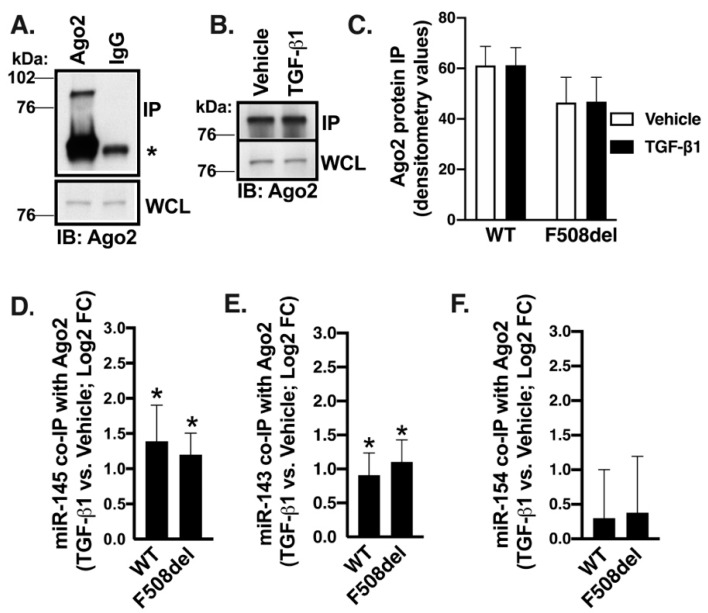
TGF-β1 selectively recruited miRNAs validated or predicted as CFTR inhibitors to RISC in WT-and F508del-CFBE cells (**A**). Representative WB images showing that Ago2 was specifically immunoprecipitated (IP) with the anti-Ago2 antibody RN003M. Non-immune IgG was used as a negative control. Ago2 protein was detected in the IP and whole cell lysate (WCL) samples with the anti-Ago2 antibody. IP Ago2 was normalized for WCL Ago2. The non-specific band of the IP antibody is marked with an asterisk. Representative WB images (**B**) and summary of data (**C**) showing that TGF-β1 had no effect on Ago2 abundance in WCL and the Ago2 IP efficiency in WT- and F508del-CFBE cells. (**D**–**F**). The co-IP of miRNAs with Ago2 IP was detected by qRT-PCR in the IP samples. Threshold cycle (Ct) values of miR-145 in vehicle-treated cells were subtracted from Ct values of miR-145 in TGF-β1-treated cells to generate ΔCts. FC in miR-145 level between samples was determined using the equation 2^−ΔC*t*^ and expressed as Log2 FC versus vehicle. TGF-β1 increased the miR-145 (**D**) and miR-143 (**E**) co-IP with Ago2 in WT-CFBE and F508del-CFBE cells and did not affect the co-IP of miR-154 (**F**). Error bars, S.E.M. *N* = 10/group. * *p* < 0.05.

**Figure 10 ijms-20-04933-f010:**
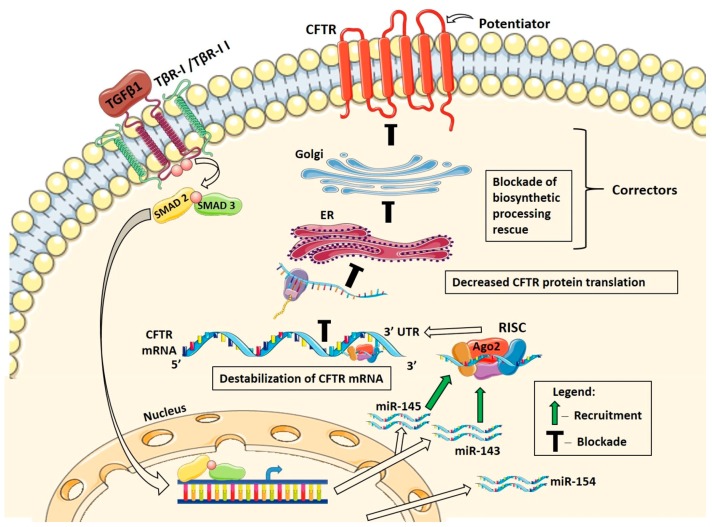
Summary of data showing TGF-β1 recruitment of miR-143 and miR-145 to RISC to destabilize CFTR mRNA and impair the corrector/potentiator-mediated rescue of F508del-CFTR. miR-143 and miR-145 are transcriptionally regulated by the canonical Smad2/3 TGF-β1 signaling pathway. Modulating the total cellular miRNA levels is not sufficient to regulate expression of CFTR. Selective RISC recruitment of the experimentally validated CFTR inhibitors, miR-143 and miR-145, by TGF-β1 destabilizes CFTR mRNA, decreases CFTR protein translation, and impairs the corrector/potentiator-mediated rescue of CFTR protein and channel function in F508del HBE cells. Predicted CFTR inhibitor, miR-154 was upregulated but not recruited to RISC by TGF-β1, and thus it is not predicted to play a role in TGF-β1-induced degradation of CFTR mRNA. ER, endoplasmic reticulum; TβR-I, TGF-β receptor I; Ago2, Argonaute2; RISC, RNA-induced silencing complex.

**Table 1 ijms-20-04933-t001:** Expression scores of miR-145, TβR-I, and TβR-II from ISH and IHC experiments, as presented in Figure 3. Two blinded reviewers assigned a score between 0 and +3 based on the miR-145 expression level in bronchial epithelium. The agreement between reviewers was 100%. The expression of miR-145 varied in F508 epithelia but was increased, compared to controls. *, image shown in Figure 3.

TISSUE	MIR-145	TβR-I	TβR-II
CONTROL-1 *	0	+2	+3
CONTROL-2	0	+2	+3
CONTROL-3	0	+2	+3
F508del-1	+3	+2	+2
F508del-2	+1	+2	+3
F508del-3 *	+2	+2	+3
COPD-1 *	+3	+1	+3
COPD-2	+3	+2	+3
IPF-1	+3	+1	+3

**Table 2 ijms-20-04933-t002:** Contingency table comparing the number of miRNAs significantly changed by TGF-β1 in WT-CFBE and F508del-CFBE cells. Data are from the small RNA-seq experiments in WT- and F508del-CFBE cells. F508del-CFTR modulated how TGF-β1 changed the miRNA landscape. The range of Log2 FC and *p* values are for the expression of individual miRNAs in the TGF-β1 versus vehicle-treated groups. *N* = 6 in triplicates from six different cultures/group. * *p* < 0.05; ** *p* < 0.01; *** *p* < 0.001; **** *p* < 0.0001 (two-sided Fisher’s exact test).

MIRNA (N)	WT-CFBE CELLS		F508DEL-CFBE CELLS	
UPREGULATED	119	Log2 FC Range 0.25 to 2.6	75	Log2 FC Range 0.38 to 2.3
		*p* Value Range * to ****		*p* Value Range * to ****
**DOWNREGULATED**	50	Log2 FC Range −1.44 to −0.2	110	Log2 FC Range −3.9 to −0.36
		*p* Value Range * to ****		*p* Value Range * to ****
**TOTAL DYSREGULATED**	169	/	185	*p* Value **** (Fisher’s exact test)

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
