# Peer review of "Transforming Growth Factor-β1 Selectively Recruits microRNAs to the RNA-Induced Silencing Complex and Degrades CFTR mRNA under Permissive Conditions in Human Bronchial Epithelial Cells"

_ijms, 2019, doi:10.3390/ijms20194933_

Round 1
Reviewer 1 Report
Mitash et al. provide a significant amount of biochemical, cell-biologcal, as wel as functional evidence to gain a molecular insight into the mechanism of CFTR channel regulation by TGF-ß1 signaling that involves miRNA-dependent silencing. The aim is to improve and optimize therapeutic strategies to combat cystic fibrosis and other lung diseases. Appropriately, the investigators use patient-derived cell lines, novel pharmacological agents, and state-of-the art methodology (e.g. RNAseq) to narrow down the group of miRNAs that are most effective at modulating CFTR physiology. The study is experimentally very well performed with sufficent amount of data to support the author's hypothesis. The manuscript is easy to read and ends with an brilliant schematic describing the derived concepts in an elegant way. The only area that needs more attention is the text wherein multiple formal mistakes and inconsistencies are found. Although these details do not change the overall meaning of the article, their correction may contribute to improved readability and general understanding of these critical findings by scientific audience.
1) The sentence on line 81 "Post-transcriptional regulation by miRNAs, as effectors of TGF-ß1, has been suggested because antagonism of the experimentally validated CFTR inhibitor miRNA-145-5p (referred to as miR-145 in the manuscript) partially blocked TGF-ß1 inhibition of corrector VX-809 rescue of F508del-CFTR in primary differentiated human bronchial epithelial (HBE) cells [24]." should read something like this: "Post-transcriptional regulation by miRNAs, as effectors of TGF-ß1, has been suggested because antagonism of miRNA-145 (referred to as miR-145 in the manuscript) by miRNA-145-5p, an experimentally validated CFTR inhibitor, partially blocked TGF-ß1 inhibition of corrector VX-809-mediated rescue of F508del-CFTR in primary differentiated human bronchial epithelial (HBE) cells [24].", since i) antagonism should not relate to an inhbitor but to its target; and ii) miR-145 stands as an abbreviation for the target and not the inhibitor.
2) Similarly, the sentence on line 86 "Yet, the degradation of F508del-CFTR mRNA by TGF-ß1 has not been directly demonstrated." is not semantically correct as it is not TGF-ß1 that would be expected to degrade F508del-CFTR mRNA through direct interaction. Please rephrase.
3) Please, mention the respective correctors used also in the legend to Figs. 5 and 7.
4) The title of Figure 3 on line 232 "miRNA-145 expression was increased in native human bronchial epithelia" could be expanded to read something like this: "miRNA-145 expression was increased in native human bronchial epithelial samples from lung disease patients" for impoved clarity.
5) The legend to Figure 4D does not mention that the statistics indicated above the graph (**,***,NS) refers to the decreased by >1Log2FC group. Please mend.
6) The graph legend to Fig. 7A,B reads "Control+Vehicle" and "Control+TGF-ß1", whereas it should read "Inh-Control+Vehicle" and "Inh-Control+TGF-ß1" according to what is stated in the text and the figure legend. Also, the "+" symbol should be printed in bold in the graph legend to Fig. 7C (Inh-Control+TGF-ß1) for improved consistency. Please revise.
7) Please, indicate unit for the y axis of Fig. 9C.
8) The statement of line 532, "the miR-145 upregulation and CFTR mRNA inhibition did not show negative correlation" suggests that CFTR mRNA inhibition was expected to decrease with increasing miR-145 levels, which is opposite to data/hypothesis. Please correct this statement.
9) Please remove "pp percentage predicted" (line 848) from the list of abbreviations or incorporate this abbreviation into the text on lines 59 and 68.
10) Add "SNP" (line 575), "IRB" (line 589), "PAP" (line 672), "AP" (line 678) to the list of abbreviations.
11) Please order all abbreviations alphabetically in the list on line 848.
There are also few corrections suggested below. Arrow => points to the correct version and the corresponding adjustment is indicated within a pair of angle brackets <>.
1) line 32: new generation => new-generation <hyphen>
2) lines 34, 304, 381, 394, 401, 496, 816, 818: TGF-ß1 induced => TGF- ß1-induced <hyphen>
3) lines 57, 63, 139: FDA approved => FDA-approved <hyphen>
4) line 57: ~ 15% => ~15% <no space>
5) line 59: Forced Expiratory Volume => forced expiratory volume <lower-case>
6) line 60: (FEV1) => (FEV1) <no subscript>
7) line 65: newest-generation correctors => newest generation correctors <no hyphen>
8) line 65: VX-445 have = VX-445, have <comma>
9) lines 72, 156, 177, 524, 628, 630, 828, 831, 832, 834, 836, 842: and => , and <comma>
10) lines 76, 93, 95, 116, 400: F508del => F508del <no italics>
11) line 77: ours suggest => ours, suggest <comma>
12) line 85: translation but => translation, but <comma>
13) lines 88, 486, 566: miRNA mediated => miRNA-mediated <hyphen>
14) line 89: recognition sequence(s), usually in the mRNA => target mRNA recognition sequence(s), usually in the <target mRNA>
15) line 91: RISC associated => RISC-associated <hyphen>
16) line 102: by the TGF-ß1-miRNA cross-talk => by aberrant TGF-ß1-miRNA cross-talk <aberrant>
17) lines 110, 396, 598: human bronchial epithelial => HBE
18) line 112: RISC, and => RISC and <no comma>
19) line 124: C002 and C18 => C18 and C002
20) lines 125, 139, 161: 5uM => 5 uM <space>
21) line 127: western blotting => western blotting (WB) <(WB)>
22) lines 130, 136, 153, 317, 326, 388, 503: corrector rescued => corrector-rescued <hyphen>
23) lines 131, 137, 139, 165, 166, 174, 310, 317, 325, 388: CFTR mediated => CFTR-mediated <hyphen>
24) line 133: Epithelial Sodium Channel => epithelial sodium channel <lower-case>
25) line 134: activator and => activator, and <comma>
26) line 135: inhibitor were => inhibitor, were <comma>
27) line 137: CFTRInh-172 => CFTRinh-172 <lower-case i>
28) lines 138, 148, 167, 372, 389: ENaC mediated => ENaC-mediated <hyphen>
29) lines 140, 474: although => , although <comma>
30) line 156: function and => function, and <comma>
31) line 157: images (A) => images (A) <only one space>
32) lines 160, 211: (B) => (B) <B in bold>
33) line 161: (HCL/BSA) => HCl/BSA <lower-case l, no parentheses>
34) lines 168, 619, 634: 1mM => 1 mM <space>
35) lines 174, 292: cell => cells <plural>
36) lines 176, 255, 383, 386, 405, 457, 714: vehicle treated = > vehicle-treated <hyphen>
37) line 177: Eight => 8
38) line 177: ; => , <comma>
39) line 178: standard error of the mean (S.E.M.). => standard error of the mean (S.E.M.). <smaller font>
40) lines 179, 211, 297, 393, 406, 822: **** p => ****, p <comma>
41) line 183: 1-kb => 1 kb <no hyphen>
42) lines 187, 201: later cells => later, cells <comma>
43) line 191: manner is => manner, is <comma>
44) line 193: Actinomycin D => actinomycin D <lower-case a>
45) lines 201, 762, 799: 24h => 24 h <space>
46) line 201: later cells => later, cells <comma>
47) lines 204, 249, 532, 558, 577, 750, 751 (2x): or => , or <comma>
48) line 205: real-time quantitative reverse transcriptase PCR => real-time quantitative reverse transcriptase PCR <smaller font>
49) line 206: at the 4 h => at 4 h <no the>
50) line 206: (t = 0, => (t = 0) <parenthesis, no comma>
51) line 207: delay => decay
52) line 207: C/C0 =e-kdt => C/C0 = e-kdt <space>
53) line 208: curves => curve <singular>
54) lines 210, 458, 806: TGF-ß1 treated => TGF-ß1-treated <hyphen>
55) line 210: (A) => (A) <A in bold>
56) lines 222, 235, 249, 252, 255, 260, 281, 283, 286, 288, 294, 299, 302, 309, 313, 317, 318, 319, 321, 323: Control => control <lower-case c>
57) line 227: Receptor => receptor <lower-case r>
58) line 227: above => above-mentioned
59) line 236: Locked Nucleic Acids (LNAs) => locked nucleic acid (LNA) <lower-case, singular>
60) line 237: simiral => similar
61) line 240: was => were <plural>
62) line 244: is was => was <no is>
63) line 244: Controls => controls <lower-case c>
64) lines 249, 252, 261, 282, 286, 292, 295, 299, 301, 303: Other => other <lower-case o>
65) line 253: baseline miR-145 => baseline of miR-145 <of>
66) line 256: tissues. (Fig. 4A versus Fig. 3A and Table 1) => tissues (Fig. 4A versus Fig. 3A and Table 1) <no period>
67) line 257: and validate => and hence validate <hence>
68) line 259: log2 => Log2 <upper-case L>
69) lines 283, 406 (4x): Log2FC => Log2 FC <space>
70) line 285: TGF-ß1 => TGF-ß1 treatment <treatment>
71) line 286: above in => above, in <comma>
72) lines 287, 848: Glyceraldehyde 3-phosphate dehydrogenase => glyceraldehyde-3-phosphate dehydrogenase <lower-case g, hyphen>
73) line 294: A-D => A-D <A and D in bold>
74) line 300: ) => ) <no bold>
75) line 301: Gontrol => control
76) line 311: demonstrate => demonstrated <past tense>
77) lines 324, 387: Ussing recordings => Ussing chamber recordings <chamber>
78) line 326: F508del-CFTR mediated => F508del-CFTR-mediated <hyphen>
79) line 328: Ten => 10
80) lines 328, 391: (A & B) => (A and B) <and>
81) line 329: (C & D) => (C and D) <and>
82) lines 355, 378: selected => select
83) line 362: Inh-control => Inh-Control <upper-case C>
84) line 366: corrector mediated => corrector-mediated <hyphen>
85) line 391: Eight => 8
86) line 392: (C-F). Error => (C-F). Error <only one space>
87) line 404: landscape. The => landscape. The <only one space>
88) line 406 (2x): exact => exact test <test>
89) line 425: A and B => A and B <A, B in bold>
90) line 428: after TGF-ß1 => after TGF-ß1 <only one space>
91) line 431: mRNA. The => mRNA. The <only one space>
92) lines 436, 450: IP => immunoprecipitated
93) lines 436, 802: western blotting => WB
94) line 439: (Figs. 9A-C) => (Figs. 9A-9C)
95) line 451: of => or
96) line 453: immunoprecipitating => IP
97) line 460: that TGF-ß1 => that (E) TGF-ß1 <(E), E in bold>
98) line 467: (Fig. 8B versus Fig. 9F) => (Fig. 8C versus Fig. 9F) <8C>
98) line 469: inhibitor, miR-154 => inhibitor miR-154 <no comma>
99) line 489: od => of
100) lines 490, 494: corrector/potentiator mediated => corrector/potentiator-mediated <hyphen>
101) line 498: complex => complex. <period>
102) line 517: patients is => patients, is <comma>
103) line 524: COPD, and => COPD and <no comma>
104) line 525: background allowed => background, allowed <comma>
105) line 536: Smad2/3 mediated => Smad2/3-mediated <hyphen>
106) line 539: RISC-associated miRNA => RISC-associated, miRNA <comma>
107) line 541: 10% of => 10%, of <comma>
108) line 545: inhibitor, miR-154 => inhibitor miR-154 <no comma>
109) line 549: mechanism(s) how => mechanism(s) of how <of>
110) line 557: disease, may => disease may <no comma>
111) line 560: miRNA and/or mRNA specific => miRNA- and/or mRNA-specific <2x hyphen>
112) line 572: disease specific => disease-specific <hyphen>
113) line 585: human bronchial epithelial HBE => HBE <no human bronchial epithelial>
114) line 587: Core => core <lower-case c>
115) line 594: Transwell => transwell <lower-case t>
116) line 595: ~ 2 => ~2 <no space>
117) line 595: at => in
118) line 596: Formalin-fixed, and => Formalin-fixed, <no and>
119) line 605: MBL => (MBL) <parentheses>
120) line 605: anti-TßR-I, => anti-TßR-I and <and>
121) lines 606, 688: AP08190PU-N, and => AP08190PU-N and <no comma>
122) line 606: from Acris => were from Acris <were>
123) lines 607, 802: horseradish peroxide-conjugated => horseradish peroxidase-conjugated <peroxidase-conjugated>
124) lines 608, 622, 626: ; => , <comma>
125) line 610: HCl+1 => HCl + 1 <2x space>
126) line 615: CFFT-002 and C18 (VRT-534) => C18 (VRT-534) and C002
127) lines 625, 657: 6-week => 6 week <no hyphen>
128) line 627: CA). The => CA). The <only one space>
129) line 627: were => was <singular>
130) line 629: basolateral => basolateral: <colon>
131) line 629: Na gluconate => sodium gluconate <sodium>
132) line 631: O2 => O2 <2 in subscript>
133) line 631: CO2 => CO2 <2 in subscript>
134) line 631: 37°C => 37 °C <space>
135) line 631: 5-minute => 5 min <min>
136) line 632: 50µM => 50 µM <space>
137) line 633: cyclic adenosine monophosphate (cAMP) => cAMP <standard abbreviation>
138) line 643: ImageJ based => ImageJ-based <hyphen>
139) line 646: activity we => activity, we <comma>
140) line 647: 1kb => 1 kb <space>
141) line 651: 1µg/well => 1 µg/well <space>
142) line 653: Fetal Bovine Serum => fetal bovine serum <lower-case>
143) line 655: (Promega), according => (Promega) according <no comma>
144) line 658: actinomycin-D => actinomycin D <no hyphen>
145) line 660: hybridization (ISH) => hybridization
146) line 662: RNase free => RNase-free <hyphen>
147) line 662: Formalin-fixed paraffin-embedded => Formalin-fixed, paraffin-embedded <comma>
148) line 662: 5µm => 5 µm <space>
149) lines 663, 669, 675: ºC => °C <°>
150) line 664: Phosphate Buffered Saline => phosphate-buffered saline <lower-case, hyphen>
151) line 665: Woburn MA => Woburn, MA <comma>
152) lines 669, 673: hr => h
153) line 670: Saline-Sodium Citrate => saline-sodium citrate <lower-case>
154) line 671: Tris-Buffered Saline => Tris-buffered saline <lower-case>
155) line 676: 2mM => 2 mM <space>
156) line 677: 50mM => 50 mM <space>
157) line 677: Tween => Tween-20
158) line 677: 100mM => 100 mM <space>
159) line 681: xylene => xylene treatment <treatment>
160) line 683: Immunohistochemistry (IHC) => Immunohistochemistry
161) line 685: Tuscon => Tucson
162) line 686: Cell Conditioning => cell conditioning <lower-case>
163) line 687: minutes => min
164) line 687: TGF-ß Receptor (TßR)-I => TßR-I
165) line 688: respectively at => respectively, at <comma>
166) line 689: DAB (3,3’-diaminobenzidine) => 3,3’-diaminobenzidine (DAB)
167) line 690: biotin free => biotin-free <hyphen>
168) line 696: Dnase treatment => DNase treatment <upper-case N>
169) line 696: Rnase-free Dnase => RNase-free DNase <2x upper-case N>
170) lines 698, 768: Nanodrop => NanoDrop <upper-case D>
171) line 699: Real-time Quantitative Reverse-Transcription PCR (qRT-PCR) => Real-time Quantitative Reverse-Transcription PCR
172) line 700: Real time => Real-time <hyphen>
173) line 701: 20ng => 20 ng <space>
174) line 701: Reverse Transcription => reverse transcription <lower-case>
175) line 701: Reverse Transcriptase => reverse transcriptase <lower-case>
176) line 709: AGGGATGATGTTCTGGAGAGCC-3’[22] => AGGGATGATGTTCTGGAGAGCC-3’ [22] <space>
177) lines 709, 741: , and => and <no comma>
178) line 712: threshold, and the => threshold, the <no and>
179) line 715: Fold change (FC) => Fold change
180) line 715: following formula => formula
181) lines 716, 807: FC= => FC = <space>
182) lines 721, 722: Thermo Fisher => Thermo Fisher Scientific <Scientific>
183) line 723: #001973), using => #001973) using <no comma>
184) line 724: uL => ul <lower-case l>
185) line 725: miRNA specific => miRNA-specific <hyphen>
186) line 727: 20uL => 20 ul <space, lower-case l>
187) line 737: 2’-O-Methyl, 2’-O-Methoxyethyl => 2’-O-methyl, 2’-O-methoxyethyl <2x lower-case m, 2x O in italics>
188) line 739: Inhibitor control => Inhibitor Control <upper-case C>
189) line 743: Exiqon) referred => Exiqon), referred <comma>
190) line 743: ontrol => control
191) line 744: added => and added <and>
192) line 747: RNA mediated => RNA-mediated <hyphen>
193) line 753: Seventy-two hours => 72 h
194) line 755: collagen coated, Transwell=> collagen-coated transwell <hyphen, no comma, lower-case t>
195) line 756: ~ 7 => ~7 <no space>
196) line 759: shSmad3 individually => shSmad3, individually <comma>
197) line 759: together depleted => together, depleted <comma>
198) line 766: 100ng => 100 ng <space>
199) line 768: Fluorometer and => Fluorometer, and <comma>
200) line 773: pre sequencing => pre-sequencing <hyphen>
201) line 775: 2.5pM => 2.5 pM <space>
202) line 775: bp => base pair (bp)
203) line 775: single read => single-read <hyphen>
204) line 779: (GRCh38) and => (GRCh38), and <comma>
205) line 780: cutadapt => Cutadapt <upper-case C>
206) line 780: base-pair (bp) => bp
207) line 781: 10bp => 10 bp <space>
208) line 781: identifiers => identifier <singular>
209) line 784: counted, and => counted and <no comma>
210) line 785: EdgeR => edgeR <lower-case e>
211) line 790: Release => release <lower-case r>
212) line 792: RISC IP (RIP) assay => RISC IP assay
213) line 795: anti-EIF2C2(Ago2) => anti-EIF2C2 (Ago2) <space>
214) line 799: lysed, and => lysed and <no comma>
215) line 801: 3h => 3 h <space>
216) line 801: 4°C => 4 °C <space>
217) line 803: goat-anti-mouse => goat anti-mouse <space>
218) line 817: expression a => expression, a <comma>
219) line 820: (GEO accession number GSE128765) the => (GEO accession number GSE128765), the <comma>
220) line 835: review & editing => review and editing <and>
221) line 837: Grants => grants <lower-case g>
222) line 843: cells, and => cells and <no comma>
223) line 844: => McKiernan, for => McKiernan for <no comma>
224) line 844: provide => provided <past tense>
225) line 845: CFFT-002 and C18 => C18 and C002
226) line 848: Cystic Fibrosis Transmembrane Regulator => cystic fibrosis transmembrane regulator <lower-case>
227) line 848: Airway Surface Liquid => airway surface liquid <lower-case>
228) line 848: Cystic Fibrosis => cystic fibrosis <lower-case>
229) line 848: Deletion of Phe508 => deletion of Phe508 <lower-case d>
230) line 848: Transforming Growth Factor => transforming growth factor <lower-case>
231) line 848: Endoplasmic Reticulum => endoplasmic reticulum <lower-case>
232) line 848: Forced Expiratory Volume => forced expiratory volume <lower-case>
233) line 848: Human Bronchial Epithelium => human bronchial epithelium <lower-case>
234) line 848: Wild Type => wild type <lower-case>
235) line 848: Bovine Serum Albumin => bovine serum albumin <lower-case>
236) line 848: Epithelial Sodium Channel => epithelial sodium channel <lower-case>
237) line 848: Cyclic adenosine monophosphate => cyclic adenosine monophosphate <lower-case c>
238) line 848: Cystic Fibrosis Foundation Therapeutics => cystic fibrosis foundation therapeutics <lower-case>
239) line 848: Complimentary DNA => complementary DNA <complementary>
240) line 848: In situ hybridization => in situ hybridization <lower-case i>
241) line 848: Locked Nucleic Acid => locked nucleic acid <lower-case>
242) line 848: Immunohistochemistry => immunohistochemistry <lower-case i>
243) line 848: Protein Kinase A => protein kinase A <lower-case>
244) line 848: Human embryonic kidney => human embryonic kidney <lower-case h>
245) line 848: Messenger RNA => messenger RNA <lower-case m>
246) line 848: Chronic Obstructive Pulmonary Disease => chronic obstructive pulmonary disease <lower-case>
247) line 848: idiopathic pulmonary fibrosis => idiopathic pulmonary fibrosis <smaller font>
Supplementary file:
1) Table S1 legend: uniquely dysregulated in WT- and F508del-CFBE cells => uniquely dysregulated in WT- or F508del-CFBE cells <or>
2) Table S1 legend: **** p => ****, p <comma>
Author Response
September 26, 2019
Ms. Lynne Liu
Assitant Editor
International Journal of Molecular Sciences
Dear Ms. Liu:
We present to you the revised manuscript entitled “Transforming Growth Factor-β1 selectively recruits microRNAs to the RNA induced silencing complex and degrades CFTR mRNA under permissive conditions in human bronchial epithelial cells” for publication at the International Journal of Molecular Sciences special issue “The Role of Non-Coding RNAs in Human Lung Health and Disease”.
We thank the reviewers for their thorough review and insightful comments and suggestions that improved the quality of our manuscript. We provide point-by-point responses.
Reviewer 1:
1) The sentence on line 81 "Post-transcriptional regulation by miRNAs, as effectors of TGF-ß1, has been suggested because antagonism of the experimentally validated CFTR inhibitor miRNA-145-5p (referred to as miR-145 in the manuscript) partially blocked TGF-ß1 inhibition of corrector VX-809 rescue of F508del-CFTR in primary differentiated human bronchial epithelial (HBE) cells [24]." should read something like this: "Post-transcriptional regulation by miRNAs, as effectors of TGF-ß1, has been suggested because antagonism of miRNA-145 (referred to as miR-145 in the manuscript) by miRNA-145-5p, an experimentally validated CFTR inhibitor, partially blocked TGF-ß1 inhibition of corrector VX-809-mediated rescue of F508del-CFTR in primary differentiated human bronchial epithelial (HBE) cells [24].", since i) antagonism should not relate to an inhbitor but to its target; and ii) miR-145 stands as an abbreviation for the target and not the inhibitor.
Response: We revised this long and complex sentence to clarify the point. We refer to miR-145-5p as miR-145 in the manuscript. We replaced the phrase ‘antagonism against miR-145-5p” with ‘specific oligonucleotides against miR-145-5p’ The revised text is:
“Post-transcriptional regulation by miRNAs, as effectors of TGF-β1, has been suggested because specific oligonucleotide against an experimentally validated CFTR inhibitor miRNA-145-5p (referred to as miR-145 in the manuscript) partially prevented TGF-β1 inhibition of the corrector-rescued F508del-CFTR in primary differentiated human bronchial epithelial (HBE) cells [24].”
2) Similarly, the sentence on line 86 "Yet, the degradation of F508del-CFTR mRNA by TGF-ß1 has not been directly demonstrated." is not semantically correct as it is not TGF-ß1 that would be expected to degrade F508del-CFTR mRNA through direct interaction. Please rephrase.
Response: We revised the text to correct the semantic error:
“If miRNAs are the mediators, it is expected that treatment with TGF-β1 should decrease CFTR mRNA level by accelerating its degradation. Yet, the degradation of F508del-CFTR mRNA after TGF-β1 treatment has not been demonstrated.”
3) Please, mention the respective correctors used also in the legend to Figs. 5 and 7.
Response: The corrector names used in the experiments in Fig. 5 and Fig. 7 were added to the legends.
4) The title of Figure 3 on line 232 "miRNA-145 expression was increased in native human bronchial epithelia" could be expanded to read something like this: "miRNA-145 expression was increased in native human bronchial epithelial samples from lung disease patients" for impoved clarity.
Response: The suggested change has been made.
5) The legend to Figure 4D does not mention that the statistics indicated above the graph (**,***,NS) refers to the decreased by >1Log2FC group. Please mend.
Response: The requested changes were made and the legend has been thoroughly revised:
“B. Proportions of the HBE cell lines, shown in A, with expression of miR-145 increased by > 1 Log2 FC or unchanged (< 1 Log2 FC).“
“D. Proportions of the HBE cell lines, shown in C, with expression of CFTR mRNA increased by > 1 Log2 FC, decreased by > 1 Log2 FC, or unchanged.”
“Error bars, S.E.M. *, p < 0.05; **, p < 0.01; ***, p < 0.001; ****, p < 0.0001 TGF-β1 versus vehicle control (A and C), proportions of cell lines with miR-145 expression > 1 Log2 FC versus unchanged (B), and proportions of cell lines with CFTR mRNA expression > 1 log2 FC versus unchanged and decreased (D).”
6) The graph legend to Fig. 7A,B reads "Control+Vehicle" and "Control+TGF-ß1", whereas it should read "Inh-Control+Vehicle" and "Inh-Control+TGF-ß1" according to what is stated in the text and the figure legend. Also, the "+" symbol should be printed in bold in the graph legend to Fig. 7C (Inh-Control+TGF-ß1) for improved consistency. Please revise.
Response: The graph legend was corrected.
7) Please, indicate unit for the y axis of Fig. 9C.
Response: The y-axis was corrected to include the units as “densitometry values”. The information about normalization of immunoprecipitated Ago2 was moved to the figure legend:
“IP Ago2 was normalized for WCL Ago2.”
8) The statement of line 532, "the miR-145 upregulation and CFTR mRNA inhibition did not show negative correlation" suggests that CFTR mRNA inhibition was expected to decrease with increasing miR-145 levels, which is opposite to data/hypothesis. Please correct this statement.
Response: We made the suggested change to correctly summarize our data. The data demonstrate that specific recruitment of miR-145 to RISC, and not the total cellular level of the miRNA, predicts its inhibitory potential for CFTR mRNA. The statement discussed by the reviewer was revised:
“Although TGF-β1 inhibited CFTR mRNA and increased miR-145 expression in HBE cells from lungs with CF, COPD, or IPF, the miR-145 upregulation and CFTR mRNA inhibition did not show correlation (Figs. 4 and 5).”
9) Please remove "pp percentage predicted" (line 848) from the list of abbreviations or incorporate this abbreviation into the text on lines 59 and 68.
Response: The requested changes were made.
10) Add "SNP" (line 575), "IRB" (line 589), "PAP" (line 672), "AP" (line 678) to the list of abbreviations.
Response: We added the abbreviations.
In regards to the PAP pen, it is a special marking pen that provides a thin film-like hydrophobic barrier when a circle is drawn around a specimen on a slide. The original name was derived from Peroxidase, Anti-Peroxidase (PAP) Method, an IHC technique used for paraffin-embedded tissues. The PAP pen however doe not have the PAP function, only the name. We reviewed many publications and it is customary to use the mane without providing the explabation because it may be misleading. We would like to ask the reviewer about an opinion whether to include or remove the explanation of the PAP abbreviation.
11) Please order all abbreviations alphabetically in the list on line 848.
Response: The abbreviations are now listed alphabetically.
There are also few corrections suggested below. Arrow => points to the correct version and the corresponding adjustment is indicated within a pair of angle brackets <>.
Response: We are especially thankful for this thorrough review. The suggested changes were made except of placement of <coma> before <or> if only two categories were compared. In addition, we thorroughly revised the entire manuscript making similar linguistic changes, when necessary to be consistent with the reviewer’s suggestions.
1) line 32: new generation => new-generation <hyphen>
2) lines 34, 304, 381, 394, 401, 496, 816, 818: TGF-ß1 induced => TGF- ß1-induced <hyphen>
3) lines 57, 63, 139: FDA approved => FDA-approved <hyphen>
4) line 57: ~ 15% => ~15% <no space>
5) line 59: Forced Expiratory Volume => forced expiratory volume <lower-case>
6) line 60: (FEV1) => (FEV1) <no subscript>
7) line 65: newest-generation correctors => newest generation correctors <no hyphen>
8) line 65: VX-445 have = VX-445, have <comma>
9) lines 72, 156, 177, 524, 628, 630, 828, 831, 832, 834, 836, 842: and => , and <comma>
10) lines 76, 93, 95, 116, 400: F508del => F508del <no italics>
11) line 77: ours suggest => ours, suggest <comma>
12) line 85: translation but => translation, but <comma>
13) lines 88, 486, 566: miRNA mediated => miRNA-mediated <hyphen>
14) line 89: recognition sequence(s), usually in the mRNA => target mRNA recognition sequence(s), usually in the <target mRNA>
15) line 91: RISC associated => RISC-associated <hyphen>
16) line 102: by the TGF-ß1-miRNA cross-talk => by aberrant TGF-ß1-miRNA cross-talk <aberrant>
17) lines 110, 396, 598: human bronchial epithelial => HBE
18) line 112: RISC, and => RISC and <no comma>
19) line 124: C002 and C18 => C18 and C002
20) lines 125, 139, 161: 5uM => 5 uM <space>
21) line 127: western blotting => western blotting (WB) <(WB)>
22) lines 130, 136, 153, 317, 326, 388, 503: corrector rescued => corrector-rescued <hyphen>
23) lines 131, 137, 139, 165, 166, 174, 310, 317, 325, 388: CFTR mediated => CFTR-mediated <hyphen>
24) line 133: Epithelial Sodium Channel => epithelial sodium channel <lower-case>
25) line 134: activator and => activator, and <comma>
26) line 135: inhibitor were => inhibitor, were <comma>
27) line 137: CFTRInh-172 => CFTRinh-172 <lower-case i>
28) lines 138, 148, 167, 372, 389: ENaC mediated => ENaC-mediated <hyphen>
29) lines 140, 474: although => , although <comma>
30) line 156: function and => function, and <comma>
31) line 157: images (A) => images (A) <only one space>
32) lines 160, 211: (B) => (B) <B in bold>
33) line 161: (HCL/BSA) => HCl/BSA <lower-case l, no parentheses>
34) lines 168, 619, 634: 1mM => 1 mM <space>
35) lines 174, 292: cell => cells <plural>
36) lines 176, 255, 383, 386, 405, 457, 714: vehicle treated = > vehicle-treated <hyphen>
37) line 177: Eight => 8
38) line 177: ; => , <comma>
39) line 178: standard error of the mean (S.E.M.). => standard error of the mean (S.E.M.). <smaller font>
40) lines 179, 211, 297, 393, 406, 822: **** p => ****, p <comma>
41) line 183: 1-kb => 1 kb <no hyphen>
42) lines 187, 201: later cells => later, cells <comma>
43) line 191: manner is => manner, is <comma>
44) line 193: Actinomycin D => actinomycin D <lower-case a>
45) lines 201, 762, 799: 24h => 24 h <space>
46) line 201: later cells => later, cells <comma>
47) lines 204, 249, 532, 558, 577, 750, 751 (2x): or => , or <comma>
48) line 205: real-time quantitative reverse transcriptase PCR => real-time quantitative reverse transcriptase PCR <smaller font>
49) line 206: at the 4 h => at 4 h <no the>
50) line 206: (t = 0, => (t = 0) <parenthesis, no comma>
51) line 207: delay => decay
52) line 207: C/C0 =e-kdt => C/C0 = e-kdt <space>
53) line 208: curves => curve <singular>
54) lines 210, 458, 806: TGF-ß1 treated => TGF-ß1-treated <hyphen>
55) line 210: (A) => (A) <A in bold>
56) lines 222, 235, 249, 252, 255, 260, 281, 283, 286, 288, 294, 299, 302, 309, 313, 317, 318, 319, 321, 323: Control => control <lower-case c>
57) line 227: Receptor => receptor <lower-case r>
58) line 227: above => above-mentioned
59) line 236: Locked Nucleic Acids (LNAs) => locked nucleic acid (LNA) <lower-case, singular>
60) line 237: simiral => similar
61) line 240: was => were <plural>
62) line 244: is was => was <no is>
63) line 244: Controls => controls <lower-case c>
64) lines 249, 252, 261, 282, 286, 292, 295, 299, 301, 303: Other => other <lower-case o>
65) line 253: baseline miR-145 => baseline of miR-145 <of>
66) line 256: tissues. (Fig. 4A versus Fig. 3A and Table 1) => tissues (Fig. 4A versus Fig. 3A and Table 1) <no period>
67) line 257: and validate => and hence validate <hence>
68) line 259: log2 => Log2 <upper-case L>
69) lines 283, 406 (4x): Log2FC => Log2 FC <space>
70) line 285: TGF-ß1 => TGF-ß1 treatment <treatment>
71) line 286: above in => above, in <comma>
72) lines 287, 848: Glyceraldehyde 3-phosphate dehydrogenase => glyceraldehyde-3-phosphate dehydrogenase <lower-case g, hyphen>
73) line 294: A-D => A-D <A and D in bold>
74) line 300: ) => ) <no bold>
75) line 301: Gontrol => control
76) line 311: demonstrate => demonstrated <past tense>
77) lines 324, 387: Ussing recordings => Ussing chamber recordings <chamber>
78) line 326: F508del-CFTR mediated => F508del-CFTR-mediated <hyphen>
79) line 328: Ten => 10
80) lines 328, 391: (A & B) => (A and B) <and>
81) line 329: (C & D) => (C and D) <and>
82) lines 355, 378: selected => select
83) line 362: Inh-control => Inh-Control <upper-case C>
84) line 366: corrector mediated => corrector-mediated <hyphen>
85) line 391: Eight => 8
86) line 392: (C-F). Error => (C-F). Error <only one space>
87) line 404: landscape. The => landscape. The <only one space>
88) line 406 (2x): exact => exact test <test>
89) line 425: A and B => A and B <A, B in bold>
90) line 428: after TGF-ß1 => after TGF-ß1 <only one space>
91) line 431: mRNA. The => mRNA. The <only one space>
92) lines 436, 450: IP => immunoprecipitated
93) lines 436, 802: western blotting => WB
94) line 439: (Figs. 9A-C) => (Figs. 9A-9C)
95) line 451: of => or
96) line 453: immunoprecipitating => IP
97) line 460: that TGF-ß1 => that (E) TGF-ß1 <(E), E in bold>
98) line 467: (Fig. 8B versus Fig. 9F) => (Fig. 8C versus Fig. 9F) <8C>
98) line 469: inhibitor, miR-154 => inhibitor miR-154 <no comma>
99) line 489: od => of
100) lines 490, 494: corrector/potentiator mediated => corrector/potentiator-mediated <hyphen>
101) line 498: complex => complex. <period>
102) line 517: patients is => patients, is <comma>
103) line 524: COPD, and => COPD and <no comma>
104) line 525: background allowed => background, allowed <comma>
105) line 536: Smad2/3 mediated => Smad2/3-mediated <hyphen>
106) line 539: RISC-associated miRNA => RISC-associated, miRNA <comma>
107) line 541: 10% of => 10%, of <comma>
108) line 545: inhibitor, miR-154 => inhibitor miR-154 <no comma>
109) line 549: mechanism(s) how => mechanism(s) of how <of>
110) line 557: disease, may => disease may <no comma>
111) line 560: miRNA and/or mRNA specific => miRNA- and/or mRNA-specific <2x hyphen>
112) line 572: disease specific => disease-specific <hyphen>
113) line 585: human bronchial epithelial HBE => HBE <no human bronchial epithelial>
114) line 587: Core => core <lower-case c>
115) line 594: Transwell => transwell <lower-case t>
116) line 595: ~ 2 => ~2 <no space>
117) line 595: at => in
118) line 596: Formalin-fixed, and => Formalin-fixed, <no and>
119) line 605: MBL => (MBL) <parentheses>
120) line 605: anti-TßR-I, => anti-TßR-I and <and>
121) lines 606, 688: AP08190PU-N, and => AP08190PU-N and <no comma>
122) line 606: from Acris => were from Acris <were>
123) lines 607, 802: horseradish peroxide-conjugated => horseradish peroxidase-conjugated <peroxidase-conjugated>
124) lines 608, 622, 626: ; => , <comma>
125) line 610: HCl+1 => HCl + 1 <2x space>
126) line 615: CFFT-002 and C18 (VRT-534) => C18 (VRT-534) and C002
127) lines 625, 657: 6-week => 6 week <no hyphen>
128) line 627: CA). The => CA). The <only one space>
129) line 627: were => was <singular>
130) line 629: basolateral => basolateral: <colon>
131) line 629: Na gluconate => sodium gluconate <sodium>
132) line 631: O2 => O2 <2 in subscript>
133) line 631: CO2 => CO2 <2 in subscript>
134) line 631: 37°C => 37 °C <space>
135) line 631: 5-minute => 5 min <min>
136) line 632: 50µM => 50 µM <space>
137) line 633: cyclic adenosine monophosphate (cAMP) => cAMP <standard abbreviation>
138) line 643: ImageJ based => ImageJ-based <hyphen>
139) line 646: activity we => activity, we <comma>
140) line 647: 1kb => 1 kb <space>
141) line 651: 1µg/well => 1 µg/well <space>
142) line 653: Fetal Bovine Serum => fetal bovine serum <lower-case>
143) line 655: (Promega), according => (Promega) according <no comma>
144) line 658: actinomycin-D => actinomycin D <no hyphen>
145) line 660: hybridization (ISH) => hybridization
146) line 662: RNase free => RNase-free <hyphen>
147) line 662: Formalin-fixed paraffin-embedded => Formalin-fixed, paraffin-embedded <comma>
148) line 662: 5µm => 5 µm <space>
149) lines 663, 669, 675: ºC => °C <°>
150) line 664: Phosphate Buffered Saline => phosphate-buffered saline <lower-case, hyphen>
151) line 665: Woburn MA => Woburn, MA <comma>
152) lines 669, 673: hr => h
153) line 670: Saline-Sodium Citrate => saline-sodium citrate <lower-case>
154) line 671: Tris-Buffered Saline => Tris-buffered saline <lower-case>
155) line 676: 2mM => 2 mM <space>
156) line 677: 50mM => 50 mM <space>
157) line 677: Tween => Tween-20
158) line 677: 100mM => 100 mM <space>
159) line 681: xylene => xylene treatment <treatment>
160) line 683: Immunohistochemistry (IHC) => Immunohistochemistry
161) line 685: Tuscon => Tucson
162) line 686: Cell Conditioning => cell conditioning <lower-case>
163) line 687: minutes => min
164) line 687: TGF-ß Receptor (TßR)-I => TßR-I
165) line 688: respectively at => respectively, at <comma>
166) line 689: DAB (3,3’-diaminobenzidine) => 3,3’-diaminobenzidine (DAB)
167) line 690: biotin free => biotin-free <hyphen>
168) line 696: Dnase treatment => DNase treatment <upper-case N>
169) line 696: Rnase-free Dnase => RNase-free DNase <2x upper-case N>
170) lines 698, 768: Nanodrop => NanoDrop <upper-case D>
171) line 699: Real-time Quantitative Reverse-Transcription PCR (qRT-PCR) => Real-time Quantitative Reverse-Transcription PCR
172) line 700: Real time => Real-time <hyphen>
173) line 701: 20ng => 20 ng <space>
174) line 701: Reverse Transcription => reverse transcription <lower-case>
175) line 701: Reverse Transcriptase => reverse transcriptase <lower-case>
176) line 709: AGGGATGATGTTCTGGAGAGCC-3’[22] => AGGGATGATGTTCTGGAGAGCC-3’ [22] <space>
177) lines 709, 741: , and => and <no comma>
178) line 712: threshold, and the => threshold, the <no and>
179) line 715: Fold change (FC) => Fold change
180) line 715: following formula => formula
181) lines 716, 807: FC= => FC = <space>
182) lines 721, 722: Thermo Fisher => Thermo Fisher Scientific <Scientific>
183) line 723: #001973), using => #001973) using <no comma>
184) line 724: uL => ul <lower-case l>
185) line 725: miRNA specific => miRNA-specific <hyphen>
186) line 727: 20uL => 20 ul <space, lower-case l>
187) line 737: 2’-O-Methyl, 2’-O-Methoxyethyl => 2’-O-methyl, 2’-O-methoxyethyl <2x lower-case m, 2x O in italics>
188) line 739: Inhibitor control => Inhibitor Control <upper-case C>
189) line 743: Exiqon) referred => Exiqon), referred <comma>
190) line 743: ontrol => control
191) line 744: added => and added <and>
192) line 747: RNA mediated => RNA-mediated <hyphen>
193) line 753: Seventy-two hours => 72 h
194) line 755: collagen coated, Transwell=> collagen-coated transwell <hyphen, no comma, lower-case t>
195) line 756: ~ 7 => ~7 <no space>
196) line 759: shSmad3 individually => shSmad3, individually <comma>
197) line 759: together depleted => together, depleted <comma>
198) line 766: 100ng => 100 ng <space>
199) line 768: Fluorometer and => Fluorometer, and <comma>
200) line 773: pre sequencing => pre-sequencing <hyphen>
201) line 775: 2.5pM => 2.5 pM <space>
202) line 775: bp => base pair (bp)
203) line 775: single read => single-read <hyphen>
204) line 779: (GRCh38) and => (GRCh38), and <comma>
205) line 780: cutadapt => Cutadapt <upper-case C>
206) line 780: base-pair (bp) => bp
207) line 781: 10bp => 10 bp <space>
208) line 781: identifiers => identifier <singular>
209) line 784: counted, and => counted and <no comma>
210) line 785: EdgeR => edgeR <lower-case e>
211) line 790: Release => release <lower-case r>
212) line 792: RISC IP (RIP) assay => RISC IP assay
213) line 795: anti-EIF2C2(Ago2) => anti-EIF2C2 (Ago2) <space>
214) line 799: lysed, and => lysed and <no comma>
215) line 801: 3h => 3 h <space>
216) line 801: 4°C => 4 °C <space>
217) line 803: goat-anti-mouse => goat anti-mouse <space>
218) line 817: expression a => expression, a <comma>
219) line 820: (GEO accession number GSE128765) the => (GEO accession number GSE128765), the <comma>
220) line 835: review & editing => review and editing <and>
221) line 837: Grants => grants <lower-case g>
222) line 843: cells, and => cells and <no comma>
223) line 844: => McKiernan, for => McKiernan for <no comma>
224) line 844: provide => provided <past tense>
225) line 845: CFFT-002 and C18 => C18 and C002
226) line 848: Cystic Fibrosis Transmembrane Regulator => cystic fibrosis transmembrane regulator <lower-case>
227) line 848: Airway Surface Liquid => airway surface liquid <lower-case>
228) line 848: Cystic Fibrosis => cystic fibrosis <lower-case>
229) line 848: Deletion of Phe508 => deletion of Phe508 <lower-case d>
230) line 848: Transforming Growth Factor => transforming growth factor <lower-case>
231) line 848: Endoplasmic Reticulum => endoplasmic reticulum <lower-case>
232) line 848: Forced Expiratory Volume => forced expiratory volume <lower-case>
233) line 848: Human Bronchial Epithelium => human bronchial epithelium <lower-case>
234) line 848: Wild Type => wild type <lower-case>
235) line 848: Bovine Serum Albumin => bovine serum albumin <lower-case>
236) line 848: Epithelial Sodium Channel => epithelial sodium channel <lower-case>
237) line 848: Cyclic adenosine monophosphate => cyclic adenosine monophosphate <lower-case c>
238) line 848: Cystic Fibrosis Foundation Therapeutics => cystic fibrosis foundation therapeutics <lower-case>
239) line 848: Complimentary DNA => complementary DNA <complementary>
240) line 848: In situ hybridization => in situ hybridization <lower-case i>
241) line 848: Locked Nucleic Acid => locked nucleic acid <lower-case>
242) line 848: Immunohistochemistry => immunohistochemistry <lower-case i>
243) line 848: Protein Kinase A => protein kinase A <lower-case>
244) line 848: Human embryonic kidney => human embryonic kidney <lower-case h>
245) line 848: Messenger RNA => messenger RNA <lower-case m>
246) line 848: Chronic Obstructive Pulmonary Disease => chronic obstructive pulmonary disease <lower-case>
247) line 848: idiopathic pulmonary fibrosis => idiopathic pulmonary fibrosis <smaller font>
Supplementary file:
1) Table S1 legend: uniquely dysregulated in WT- and F508del-CFBE cells => uniquely dysregulated in WT- or F508del-CFBE cells <or>
2) Table S1 legend: **** p => ****, p <comma>
Sincerely,
Agnieszka Swiatecka-Urban, M.D.
Reviewer 2 Report
Mitash and collaborators have afforded a complicate analysis of TGF-β negative effect on mutant CFTR rescue bay correctors. The work is very well done, and conclusions are appropriate, opening to new questions, inviting to further investigate this issue. The text must be further revised to render it more clear for the reader.
I have a criticism on the use of the term "block", included also in the tittle. It is not correct to say that "TGF-β1 blocks the corrector VX-809-mediated functional rescue of F508del-CFTR". This sentence could be interpreted as a direct action of TGF-β on the corrector. The lack of correction is due to a lower amount of CFTR because TGF-β1 reduced the amount of CFTR mRNA, but the remaining CFTR can be corrected by lumacaftor. I should suggest tu use the term "reduce" instead.
It is important to notice that the number of experiments (and replicates) on each condition are not reported for most experiments. It impedes to validate the goodness of the statistical analysis.
Specific comments:
line 118 ans successive: 2.1 TGF-β1 blocks rescue of F508del-CFTR and ASL volume by the newer generation correctors
According to several recent papers, synergism of the correctors should be caused by different correction mechanisms, classifying the correctors in three classes. Thus, data on the C18 and C002 correctors used separately should be shown.
lines 151-155 and figure 1. It is not clear the meaning of the ASL variation (∆ASL). What is the control value? Is it the untreated epithelium? In any case, it remains to show the effect of TGF-β1 itself on the ASL volume regulation. If TGF-β1 reduce the ENaC activity, but it seems that TGF-β1 increases the fluid reabsortion, perhaps even in the presence of some active CFTR. Appropriate controls have to be presented, and this point have to be further discussed.
Figure 1B. I suggest to express the % change of the bands B and C normalised by the A+B sum.
Figure 1C. Traces of untreated or vehicle treated cells should be also shown.
line 180 and successive: 2.2 TGF-β1 facilitates degradation of CFTR mRNA
Does the construct pCFTR-pLuc included the mutation F508del?
Decay curves in Figure 2B indicates that "remaining mRNA" at zero time was more than 100% (112% and 123%). Thus, it is not clear what is the reference point, that is, the initial amount of mRNA, that should be 100%.
The quantification of mRNA is an absolute value that depends on the number of cells, and to the mRNA content per cell. Without an appropriate reference, these values are not quantitative. It is a good practice to normalise the qRT-PCR data by a housekeeper mRNA. It would also provide the initial CFTR-mRNA value.
line 254: If you say "Although not statistically significant", you cannot affirm that these data is important. Perhaps an adequate statistical test for variances could tell you that distributions are different, but such statistical tests must be done. Notice that graphics are logarithmic, and therefore the statistics must take in consideration it.
Figure 2E: The Pearson correlation coefficient is designed for data points with an homogeneous standard deviation. Data there is expressed as logarithm, and therefore, the standard deviations are not homogeneous. If the lineal fitting was done without including the data weights, results are invalid.
Figure 7F. It is strange that CFTR and ENaC are differently expressed, but trans-epithelial resistance (TER) have not changed. Perhaps the trans-epithelial potentials differences are changed. It should be explained.
Minor comment:
Line 47- the codon should refer to DNA sequence, not to protein number.
Author Response
September 26, 2019
Ms. Lynne Liu
Assitant Editor
International Journal of Molecular Sciences
Dear Ms. Liu:
We present to you the revised manuscript entitled “Transforming Growth Factor-β1 selectively recruits microRNAs to the RNA induced silencing complex and degrades CFTR mRNA under permissive conditions in human bronchial epithelial cells” for publication at the International Journal of Molecular Sciences special issue “The Role of Non-Coding RNAs in Human Lung Health and Disease”.
We thank the reviewers for their thorough review and insightful comments and suggestions that improved the quality of our manuscript. We provide point-by-point responses.
Reviewer 2:
Mitash and collaborators have afforded a complicate analysis of TGF-β negative effect on mutant CFTR rescue bay correctors. The work is very well done, and conclusions are appropriate, opening to new questions, inviting to further investigate this issue. The text must be further revised to render it more clear for the reader.
I have a criticism on the use of the term "block", included also in the tittle. It is not correct to say that "TGF-β1 blocks the corrector VX-809-mediated functional rescue of F508del-CFTR". This sentence could be interpreted as a direct action of TGF-β on the corrector. The lack of correction is due to a lower amount of CFTR because TGF-β1 reduced the amount of CFTR mRNA, but the remaining CFTR can be corrected by lumacaftor. I should suggest tu use the term "reduce" instead.
Response: We removed the word “block” from the description of the TGF-β1 effect on corrector mediated rescue in the entire manuscript. We replaced it with “inhibit” or “compromise” to avoid the implication that of the nature of the negative effect of TGF-β1 on CFTR correctors. We did not change the manuscript title because it does not contain the word “block”. We also used the term “impede” to describe the TGF-β1 effect in several sentences throughout the manuscript.
It is important to notice that the number of experiments (and replicates) on each condition are not reported for most experiments. It impedes to validate the goodness of the statistical analysis.
Response: We reviewed the figure legends and found that the information about number of experiments was missing from the legend for Figure 6. We thank the reviever for the commend. The information was added. All other figure legends list the number of experiments and replicates. The information is not always at the end of the legend as most figures are complex and show different assays, and thus, the information is included throughout. Whenever possible, we moved the information to the end of the legend to make it more visible (Figs. 1, 5, and 8). Since Table 2, summarizing some data from Figure 8, is shown just before Figure 8, we added the information on number of experiments used to generate Table 2 in the legend as well.
Specific comments:
line 118 ans successive: 2.1 TGF-β1 blocks rescue of F508del-CFTR and ASL volume by the newer generation correctors
According to several recent papers, synergism of the correctors should be caused by different correction mechanisms, classifying the correctors in three classes. Thus, data on the C18 and C002 correctors used separately should be shown.
Response: Data on the effect of corrector C18 and C002 used separately has been already published (ref # 43: Holleran, J. P.; Glover, M. L.; Peters, K. W.; Bertrand, C. A.; Watkins, S. C.; Jarvik, J. W.; Frizzell, R. A., Pharmacological rescue of the mutant cystic fibrosis transmembrane conductance regulator (CFTR) detected by use of a novel fluorescence platform. Mol Med 2012, 18, 685-96). We summarized the conclusions in the Results, section 2.1 to justify the combined use of corrector C18 and C002:
“the combined use of corrector C18 and CFFT-002 (C002) had a superior effect on F508del-CFTR rescue in vitro, compared to a single administration of either corrector or other small molecules [43].”
We aimed to examine TGF-β1 effects on F508-del CFTR rescued by the correctors with the strongest in vitro effect. Therefore, we used the pair of correctors.
lines 151-155 and figure 1. It is not clear the meaning of the ASL variation (∆ASL). What is the control value? Is it the untreated epithelium? In any case, it remains to show the effect of TGF-β1 itself on the ASL volume regulation. If TGF-β1 reduce the ENaC activity, but it seems that TGF-β1 increases the fluid reabsortion, perhaps even in the presence of some active CFTR. Appropriate controls have to be presented, and this point have to be further discussed.
Response: Detailed explanation of the experimental procedure and was added in the Results section:
“…we examined how TGF-β1 affects the corrector-rescue of ASL in F508del HBE cells by meniscus scanning [50]. ASL volume in ALI cultures was measured by scanning the apical meniscus before treatment. Cells were treated with correctors C18/C002 and scanned 24 h later. Subsequently, cells were treated with TGF-β1 or vehicle in the presence of correctors C18/C002 and scanned again 24 h later.“
Figure 1 legend:
“H. Summary of ASL volume measurements. ASL volume in ALI cultures of F508del HBE cells was measured by scanning the apical meniscus before treatment. Cells were scannel, treated with correctors C18/C002 for 24 h, and scanned again. Subsequently, cells were treated with TGF-β1 or vehicle in the presence of correctors C18/C002 and scanned again 24 h later. D ASL represents the change from baseline in ASL volume at 24 h after corrector rescue or at 48 h after corrector rescue with TGF-β1 of vehicle treatment..”
The discussion of the results is in Results, section 2.1 (line 148) and in the Discussion (line 498).
Figure 1B. I suggest to express the % change of the bands B and C normalised by the A+B sum.
Response: Figure 1B shows total cellular abundance of CFTR protein band B and band C. TGF-β1 decreased CFTR band B and band C abundance. Ezrin was used as a loading control to normalize CFTR expression. We do not understand the reviewer’s suggestion to normalized band B and C by the A+B sum.
For clarity, we extended the information in the Figure 1 legend about use of ezrin as a loading control to inform that it was used to normalized CFTR expression:
“Ezrin was used as a loading control to normalize CFTR expression.”
Figure 1C. Traces of untreated or vehicle treated cells should be also shown.
Response: The requested traces were added in Figure 1C.
line 180 and successive: 2.2 TGF-β1 facilitates degradation of CFTR mRNA
Does the construct pCFTR-pLuc included the mutation F508del?
Response: the pCFTR-pLuc construct contains the CFTR promoter region and not the coding sequence. The information is in the Results, section 2.2, Figure 2A legend, and Methods, section 4.6.
Decay curves in Figure 2B indicates that "remaining mRNA" at zero time was more than 100% (112% and 123%). Thus, it is not clear what is the reference point, that is, the initial amount of mRNA, that should be 100%.
Response: To clarify the mRNA degradation measurements and answer reviewer’s questions, we expanded the Methods section 4.7:
“CFTR mRNA decay was measured in 6 week ALI cultures of F508del HBE cells in the presence of the transcriptional inhibitor ActD [51, 52]. The stock solution of ActD, dissolved in DMSO, was used at a concentration of 5 µg/ml in cell culture medium with final DMSO concentration 0.1%. Total RNA was isolated 4, 8, and 24 h later, and CFTR mRNA was measured by qRT-PCR, using the relative quantitation method, as described in section 4.11 below. 18S was used as a reference gene because there was no degradation of its transcripts even after the 24-h ActD treatment (data not shown). Data were first plotted as % mRNA remaining, normalized to 18S versus time, with the first timepoint (4 h) set to 100%. Data were normalized to the 4 h timepoint to allow time for ActD to act, eliminate any potential DMSO effects, avoid initial feedback-driven transcriptional enhancement, and allow for initial poly-A tail deadenylation that precedes the first-order degradation kinetics. CFTR mRNA half-lives were calculated from the exponential decay model, based on trend line equation C/C0 = e-kdt (where C and C0 are mRNA amounts at the time t and t0, respectively, and kd is the mRNA decay constant). “
The quantification of mRNA is an absolute value that depends on the number of cells, and to the mRNA content per cell. Without an appropriate reference, these values are not quantitative. It is a good practice to normalise the qRT-PCR data by a housekeeper mRNA. It would also provide the initial CFTR-mRNA value.
Response: The information about reference gene GAPDH for CFTR mRNA expression and U6 snRNA for miRNA expression is included in the relevant methods sections and figure legends. The amount of miRNA-143, miR-145, and miR-154 co-immunoprecipitated with Ago2 could not be normalized for a reference gene because miRNAs are selectively recruited to the RISC complex (Fig.9).
We added the following information about 18S, used as reference gene in RNA degradation experiments, in Figure 2 legend:
“Total RNA was isolated and CFTR mRNA level at each time point was calculated by real-time quantitative reverse transcriptase PCR (qRT-PCR), using the relative quantitation method, with 18S as reference gene.”
We also added the primer sequences for 18S in the Methods section 4.11 Real-time Quantitative Reverse-Transcription PCR.
line 254: If you say "Although not statistically significant", you cannot affirm that these data is important. Perhaps an adequate statistical test for variances could tell you that distributions are different, but such statistical tests must be done. Notice that graphics are logarithmic, and therefore the statistics must take in consideration it.
Response: The statement was removed.
Figure 2E: The Pearson correlation coefficient is designed for data points with an homogeneous standard deviation. Data there is expressed as logarithm, and therefore, the standard deviations are not homogeneous. If the lineal fitting was done without including the data weights, results are invalid.
Response: We thank the reviewer for pointing out that Pearson correlation is not suitable statistics for these data. We assume the reviewer is referring to Figure 4E. We have reanalyzed the data using Spearman correlation and the fold change (FC) values were used instead of log transformed values. We cannot include data weights for lineal fitting because it would create more variability is a small sample size. The Methods section 4.18 was updated with information on Spearman statistics.
Figure 7F. It is strange that CFTR and ENaC are differently expressed, but trans-epithelial resistance (TER) have not changed. Perhaps the trans-epithelial potentials differences are changed. It should be explained.
Response: TER measurement was used to determine the monolayer integrity before experiments and was measured after cells were placed in the Ussing chamber and equilibrated and before addition of drugs to inhibit ENaC or activate CFTR. Please, see revised Figure 5 legend:
“The experimental conditions did not change the transepithelial resistance (TER) across the cell monolayers, measured after equilibration of cells in Ussing chamber and before addition of drugs.”
Minor comment:
Line 47- the codon should refer to DNA sequence, not to protein number.
Response: The change has been made and the revised sentence is:
“Deletion of the codon for phenylalanine at position 508 (F508del) is the most common CF-associated mutation, present in almost 90% of CF patients.
Sincerely,
Agnieszka Swiatecka-Urban, M.D.